# FreqMixAttNet: Contrastively Supervised Frequency-Mixing Attention for Time-Series Forecasting

## Abstract

Time series forecasting has gained significant attention due to its wide applicability in domains such as traffic prediction and weather monitoring. However, it remains a challenging task because of complex temporal patterns, such as multiscale periodicities and dynamic fluctuations. Existing methods often focus on either time-domain decomposition or frequency-domain analysis, but rarely integrate both effectively.In this paper, we propose FreqMixAttNet, a novel cross-domain forecasting framework that unifies time and frequency representations via a domain-mixing attention mechanism. We first introduce an adaptive convolutional wavelet decomposition to model and separate trend and seasonal components more efficiently. The seasonal part is dual-encoded in both time and frequency domains, which are treated as distinct modalities and fused through a cross-transform attention module. Meanwhile, the trend component is captured by a simple multi-scale MLP in the time domain.To further enhance robustness without pretraining, we incorporate a contrastive auxiliary loss. The combination of adaptive convolution, cross-domain mixing attention, and contrastive learning contributes to the superior performance of our method. Extensive experiments on multiple real-world benchmarks show that FreqMixAttNet consistently outperforms prior state-of-the-art methods, demonstrating the effectiveness of our unified cross-domain design.

## 1 Introduction

Time series forecasting has gained significant attention due to its applications in traffic prediction, weather forecasting, financial analysis, and energy management. However, real-world time series often exhibit complex patterns with varying fluctuations and periodic behaviors, making accurate forecasting challenging.

Recently, deep learning techniques, including CNN, Transformer, and MLP-based models, have been developed to improve forecasting. Among these, decomposition-based methods are common; they separate the input series into trend and seasonal components using moving averages across multiple scales, enabling models to capture temporal variations from coarse to fine resolutions. However, such decomposition is often too simplistic, making it difficult to properly disentangle the seasonal and trend components.

To better capture periodicity, frequency-domain methods have gained popularity. By applying transformations like the Fast Fourier Transform (FFT) or wavelet transforms, periodic structures become more explicit, with different frequencies corresponding to different cycle lengths. These methods are effective for capturing seasonal fluctuations but struggle with slowly varying long-term trends. More importantly, most focus solely on the frequency domain, without leveraging the complementary strengths of temporal-domain modeling. Recent works combine MLP-based models for trends in the time domain with filter nets for seasonality in the frequency domain ATFNet Ye & Gao (2024) TFDNetLuo et al. (2025), but they often treat the components independently, lacking effective interaction between the two domains.

A key limitation of existing methods is that temporal and frequency modeling are often treated in isolation. Time-domain features capture fine-grained variations, while frequency-domain features

highlight periodic structures; without interaction, each view remains incomplete. Treating them as two complementary modalities and enabling effective information exchange is crucial for improving forecasting performance.

Robustness is a critical challenge in time series forecasting, as real-world datasets are often noisy and limited in size, making deep models highly prone to overfitting. While pretraining can alleviate this, it typically depends on complex two-stage pipelines that hinder efficiency and practical deployment.

To address these challenges, we present **FreqMixAttNet**, a forecasting framework that unifies time- and frequency-domain representations via cross-domain mixing attention. To the best of our knowledge, this is the first work in time series forecasting to explicitly bridge time- and frequency-based models through cross-attention.

First, we introduce an adaptive convolution with weights generated from the input itself. Next, a wavelet-based decomposition splits the input into trend and seasonal components. To enable seamless knowledge transfer, we design a cross-domain attention mechanism that integrates time- and frequency-domain information for the seasonal part, while modeling the trend with an MLP in the time domain. Additionally, we incorporate a contrastive auxiliary loss into the end-to-end training process, enhancing robustness and reducing overfitting without requiring pretraining.

The main contributions of this paper are summarized as follows:

**Patch Contextual-based Adaptive Convolution**: We replace the standard Conv1D with a novel patch-based adaptive convolution module that dynamically generates convolutional weights from the input context. This design allows the convolution operation to adapt to varying inputs, enabling context-aware feature extraction—a key innovation over fixed-weight convolutions.

**Cross-domain Mixing Attention**: The seasonal component of the series is modeled separately in the time and frequency domains, which are treated as distinct modalities. We introduce a novel cross-domain Transformer to integrate time- and frequency-domain representations. By enabling direct interaction between the two domains, our model effectively captures periodic variations across modalities and substantially improves forecasting performance.

**Contrastive Auxiliary Learning**: We propose a frequency-domain contrastive learning strategy that operates in an end-to-end manner. Augmented sequences are generated via frequency mixing and jitter operations, with contrastive loss applied to enhance robustness and mitigate overfitting without pretraining. In addition, we introduce a pairwise adversarial loss to guide augmentation, further improving representation learning.

The combination of adaptive convolution, cross-domain mixing attention, and contrastive learning results in superior performance. Extensive experiments on real-world benchmarks show that FreqMixAttNet outperforms existing methods across various forecasting tasks. The code is anonymously available at https://github.com/FreqMixAttNet/FreqMixAttNet to ensure reproducibility while preserving author anonymity.

## 2 RELATE WORK

### 2.1 SERIES MODELING IN TIME DOMAIN

Recently, numerous time series forecasting algorithms have been proposed to model temporal dynamics. Transformer-based models, such as Autoformer Nie et al. (2023a), Informer Zhou et al. (2021), Crossformer Zhang & Yan (2023), PatchTSTNie et al. (2023b) and iTransformer Liu et al. (2024), CATS Lu et al. (2024) effectively capture channel-wise or temporal dependencies.

CNN-based models have also gained popularity for extracting local temporal patterns through convolutional operations, as seen in methods like MICN Wang et al. (2023), TIMESNETWu et al. (2023),TCN Luo & Wang (2024) and SCINet Liu et al. (2022).

More recently, lightweight MLP-based models have emerged, with DLinear Zeng et al. (2023) achieving competitive performance through simple linear projections. Following this, MLP-Mixer-based models like TiDE Das et al. (2023), MTS-Mixer Li et al. (2023), TSMixer Ekambaram et al. (2023) and HDMixer Huang et al. (2024) have gained attention.

To better capture periodic and trend components, series decomposition and multi-scale modeling techniques have been introduced. Autoformer Wu et al. (2021) incorporates decomposition, separating the input into seasonal and trend components using a moving average. TimeMixer Wang et al. (2024a),AMD Hu et al. (2025) and TimeMixer++ Wang et al. (2025) extend this by leveraging multi-scale representations through max pooling or convolutions to improve modeling capacity.

However, these methods operate solely in the time domain. We argue that the frequency domain provides a more effective way to capture the seasonal characteristics of time series data.

## 2.2 Freq Domain Based Time Series Forcasting

In the frequency domain, periodic patterns are modeled with transforms such as FFT or Wavelets. FEDformer Zhou et al. (2022) applies Fourier transforms before patching, while FreTS Yi et al. (2023b) employs frequency-domain MLPs. FilterNet Yi et al. (2024) uses frequency filters, and FITS Xu et al. (2024) and Affirm Wu et al. (2025) combine low- and high-pass filters. FAN Fan et al. (2024) introduces frequency-adaptive normalization, FourierGNN Yi et al. (2023a) performs GNNs in the frequency domain, and FreDF Zhang et al. (2024) employs band-specific filters. SimpleTM Chen et al. (2025) proposes inner and outer transformers over frequency coefficients.

## 2.3 Pretrain with Contract Loss

Recent advances in contrastive learning for time series have shown strong potential in improving representation learning. TS-TCC Eldele et al. (2021), TimesURL Liu & Chen (2024), AutoTCL Zheng et al. (2024) and TF-C Zhang et al. (2022) capture invariances across augmentations by aligning encoder outputs of the full series. CATCC Eldele et al. (2023) extends TS-TCC with pseudo-labels in a semi-supervised framework, while StatioCL Zheng et al. (2024) contrasts stationary and non-stationary samples within a batch. InfoTS Luo et al. (2023) employs meta-learning to adaptively choose augmentation strategies.

Most prior methods operate at the full-series level, while recent patch-based approaches such as PITS Lee et al. (2024), TS-GAC Wang et al. (2024b), and PPT Kim et al. (2025) focus on reconstructing patches to capture internal structure. However, they typically rely on two-stage pipelines—contrastive pretraining followed by MLP finetuning—which cannot be optimized jointly.

## 3 FreqMixAttNet

The model architecture (Figure 1) begins with adaptive convolution to obtain multi-scale representations. Wavelet decomposition then separates each scale into trend and seasonal components, which are modeled independently and fused via cross-attention. A contrastive auxiliary loss is further introduced to enhance end-to-end training.

### 3.1 Context-based Adaptive Convolutional Network

In multivariate time series forecasting, the historical sequence is represented within a lookback window of length $S$ with $C$ variables: $\boldsymbol{x} = x_1,...,x_S \in \mathbb{R}^{C \times S}$. Our model treats each variable independently. The task is to predict the next $L$ steps: $\boldsymbol{y} = y_{S+1},...,y_{S+L} \in \mathbb{R}^{C \times L}$.

We downsample $\boldsymbol{x}$ into $M$ scales via average pooling, yielding multi-scale sequences = $X^1,...,X^M$, where $X^m \in \mathbb{R}^{C \times S^m}$ and $S^m = S/2^m$ for the $m$-th scale.

Each univariate series $X^m$ at a given scale is divided into patches $X_p^m \in \mathbb{R}^{C \times N \times P}$, where $P$ is the patch size and $N$ is the number of patches. We propose an adaptive Convolutional mechanism that transforms each point within a patch into a $D$-dimensional representation $\overline{X}_{p,A}^m \in \mathbb{R}^{C \times N \times P \times D}$. The adaptive convolution is applied via element-wise multiplication between each point and its corresponding adaptive weights $w_a$

$$\overline{X}_{p,A}^m = \text{POOL}(X_{p,E}^m * w_a) \tag{1}$$

where $X_{p,E}^m \in R^{C*N*P*D}$ is the original $\overline{X}_p^m$ extended by duplicating its $P$ dimension $D$ times POOL is a pooling operation is used along the patch dimension $P$.

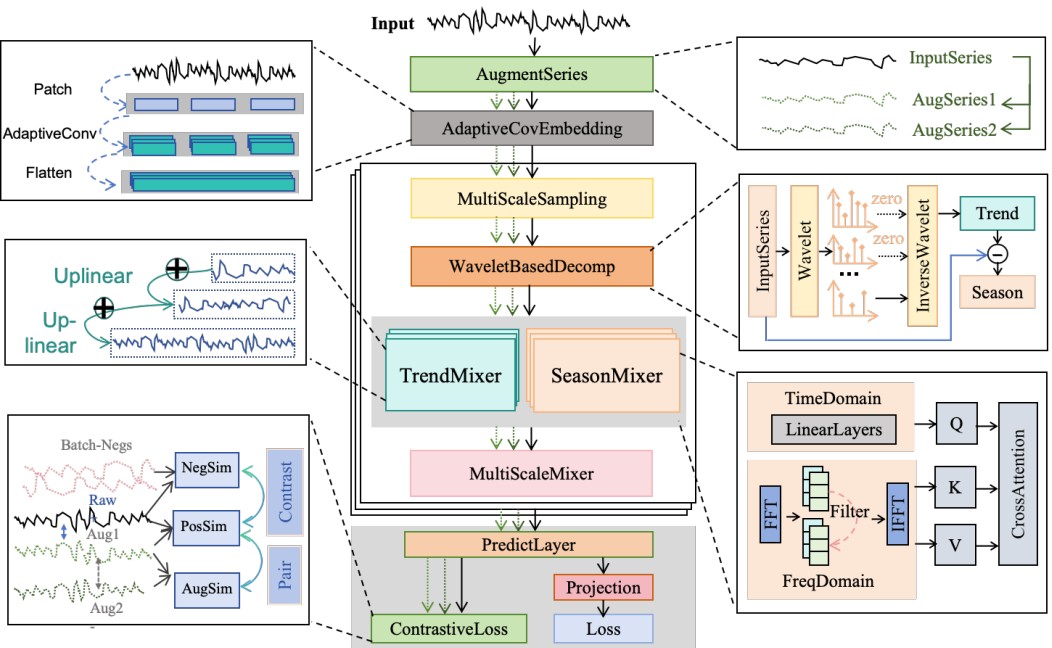

Figure 1: Overall architecture of FreqMixAttNet.

Instead of initializing $w_a$ randomly and fixing it after training, we design a context-based weight generator that learns from the input data. Specifically, we use a two-layer neural network to generate adaptive weights based on the input patches:

$$w_1 = \text{MLP}_1(X_p^m) \in \mathbb{R}^{C \times N \times P}$$
$$w_2 = \text{MLP}_2(w_1) \in \mathbb{R}^{C \times N \times P \cdot D} \quad (2)$$

Then $w_2$ is Flattened to $w_a \in \mathbb{R}^{C \times N \times P \times D}$. And the $N \times P$ structure is merged back into a sequence of length $S^m$:

$$\overline{X}^m = \text{MERGE}(X_{p,A}^m) \in \mathbb{R}^{C \times S^m \times D} \quad (3)$$

Then, we introduce a wavelet-based decomposition to separate the trend and seasonal components of the series. Specifically, the Wavelet Transform is performed on $\overline{X}^m$ to obtain high-$c^h$ and low-frequency $c^l$ coefficients and trend part is obtained by performing Inverse Wavelet Transform only on the low-frequency coefficients $c^l$:

$$c^l, c_1^h, ..., c_J^h = \text{Wavelet}(\overline{X}^m)$$
$$\overline{X}_{trend}^m = \text{IWavelet}(c^l, [0, ..., 0]) \quad (4)$$
$$\overline{X}_{season}^m = \overline{X}^m - \overline{X}_{trend}^m$$

The seasonal component is then obtained by subtracting the trend from the original sequence.

## 3.2 FREQMIXATTENTION

After decomposing the series into trend and seasonal components, we design specialized encoders for each. The trend is modeled by an MLP across multiple scales, while the seasonal part is represented jointly in frequency and time domains through FFT and a temporal MLP. To unify these perspectives, we introduce a novel cross-domain attention mechanism that explicitly fuses time–frequency information, enabling richer seasonal representations.

### 3.2.1 MLP FOR THE TREND PART

The trend component mainly reflects long-term, non-periodic variations, which are better modeled in the time domain. We therefore adopt a simple MLP with a coarse-to-fine design, as in

TimeMixerWang et al. (2024a) , to effectively encode trend features.

$$X_{trend}^m = \overline{X}_{trend}^m + \text{BottomToUp\_Mlp}(\overline{X}_{trend}^{m-1}) \tag{5}$$

### 3.2.2 Frequency Domain Modeling for the Seasonal Part

We transform the seasonal component into the frequency domain using the Discrete Fourier Transform (DFT). The DFT Oppenheim et al. (1999) maps a real-valued signal $X \in \mathbb{R}^S$ to a complex spectrum $\mathcal{F} \in \mathbb{C}^S$. For $k = 0, \ldots, S-1$, the DFT and inverse DFT (IDFT) are defined as:

$$DFT = \mathcal{F}(k) = \sum_{n=0}^{S-1} x(n) \cdot e^{-j\frac{2\pi}{S}kn}$$

$$IDFT = x(n) = \frac{1}{N} \sum_{k=0}^{N-1} \mathcal{F}(k) \cdot e^{j\frac{2\pi}{N}kn} \tag{6}$$

We apply the DFT along the temporal dimension of the seasonal series $X_{season}^m \in \mathbb{R}^{C \times D \times S^m}$ (where $S^m = \frac{S}{2^m}$ for brevity), obtaining its complex-valued frequency spectrum:

$$\mathcal{F}(\overline{X}_{season}^m) = \text{DFT}(\overline{X}_{season}^m) \tag{7}$$

The convolution theorem Katznelson (2004) states that the Fourier transform of a convolution equals the point-wise product of the individual Fourier transforms. Leveraging this property, frequency-domain filtering is introduced for time series modeling:

$$X_{season,\mathcal{F}}^m = \text{IDFT}(\mathcal{F}(\overline{X}_{season}^m) \cdot H_{filter}) \tag{8}$$

Here, $H_{filter}$ is a randomly initialized, learnable weight of length $\lceil \frac{S^m}{2} + 1 \rceil$. Unlike Yi et al. (2024), our method applies frequency filtering only to the seasonal component rather than the entire series. Since periodic patterns are better captured in the frequency domain, restricting filtering to the seasonal part enables more precise modeling.

### 3.2.3 FreqMixAttention for the Seasonal Part

In recent years, cross-attention has gained traction in multimodal fusion for its ability to integrate information across modalities. Here, we view the frequency and time domains as two complementary modalities: the frequency domain captures periodic structures of the seasonal component, while the time domain models its local fluctuations.

We model the seasonal part in the time domain with a one-layer MLP:

$$X_{season,\mathcal{T}}^m = \text{MLP}(\overline{X}_{season}^m) \tag{9}$$

The MLP operates along the temporal axis, with both input and output in $\mathbb{R}^{S^m}$.

We propose a novel cross-mixing attention mechanism that explicitly bridges time- and frequency-domain representations. Unlike conventional attention, our design treats frequency features as keys and values and time-domain features as queries, enabling the model to selectively retrieve frequency information under temporal guidance. This cross-domain interaction provides a new way to integrate periodic and local patterns for seasonal modeling.

To align both representations, we apply linear projections to each via dense layers:

$$K_{\mathcal{F}}, V_{\mathcal{F}} = X_{season,\mathcal{F}}^m W_v \in \mathbb{R}^{C \times D \times L_m}$$

$$Q_{\mathcal{T}} = X_{season,\mathcal{T}}^m W_q \in \mathbb{R}^{C \times D \times L_m} \tag{10}$$

Attention weights are computed by the scaled dot product of $Q_{\mathcal{T}}$ and $K_{\mathcal{F}}$, measuring time–frequency similarity. These weights reweight $V_{\mathcal{F}}$ to produce fused features. The cross-attention is implemented with multi-head attention:

$$X_{season,\mathcal{C}}^m = \text{CrossAttn}(X_{season,\mathcal{F}}^m) = \text{Concat}(head_1, ..., head_i, ..., head_k)W_0 \tag{11}$$

Each attention head is computed as:

$$head_i = \text{Softmax}\left(Q_{\mathcal{T}} K_{\mathcal{F}}^\top / \sqrt{d_k}\right) V_{\mathcal{F}} \tag{12}$$

Here, $d_k$ is the dimensionality of the key vectors to scale the dot product and stabilize gradients.

### 3.2.4 PREDICTION AND LOSS

The final prediction is obtained by aggregating the trend and seasonal outputs across all scales.

$$\hat{Y} = \sum_{m=1}^{M} \text{Predictor}(X_{\text{season},\mathcal{C}}^{m} + X_{\text{trend}}^{m}) \tag{13}$$

Predictor($\cdot$) refers to a single-layer MLP mapping from $S^m$ to $L$ along the temporal dimension.

The loss is the MAE between predictions and ground truth in both time and frequency domains.

$$\mathcal{L} = \frac{1}{N} \sum_{i=1}^{N} |Y_i - \hat{Y}_i| + \alpha \cdot \frac{1}{N} \sum_{i=1}^{N} |\text{DFT}(Y_i) - \text{DFT}(\hat{Y}_i)| \tag{14}$$

where DFT($\cdot$) denotes the Discrete Fourier Transform.

### 3.2.5 CONTRASTIVE AUXILIARY LOSS

In the encoder, the number of parameters is often comparable to the dataset size, and combined with noisy inputs, this makes the model susceptible to overfitting. To mitigate this, we introduce a contrastive auxiliary loss. Unlike conventional two-stage pretraining and finetuning, our method integrates contrastive learning directly into the main training process, where it is jointly optimized with the forecasting objective. This design improves robustness while avoiding the complexity of separate training stages.

To construct contrastive pairs, we apply frequency-domain jittering and create an additional augmentation by mixing the frequency components with another sample in the batch: (details in Appendix A.2)

$$X_E^{m,a1}, X_E^{m,a2} = \text{Encode}(\text{Jitter}(X^m)), \text{Encode}(\text{Mix}(X^m)) \tag{15}$$

Here, $X_E^{m,a1}$ and $X_E^m$ denote the encoded outputs of the augmented and original series at scale $m$, respectively. All scale-specific outputs are concatenated to form $h^{a1} = \text{concat}[X_E^{1,a1}, \ldots, X_E^{M,a1}]$ and $h = \text{concat}[X_E^1, \ldots, X_E^M]$. The pair $(h, h^{a1})$ serves as a positive sample, while all other samples in the batch act as negatives. The contrastive auxiliary loss is defined as:

$$\mathcal{L}^{\text{auxiliary}} = \sum_{i=1}^{B} \frac{\exp(\text{sim}(\mathbf{h}i, \mathbf{h}i^{a1})/\tau)}{\sum_{j=1,j\neq i}^{B} \exp(\text{sim}(\mathbf{h}i, \mathbf{h}j^{a1})/\tau) + \exp(\text{sim}(\mathbf{h}i, \mathbf{h}i^{a1})/\tau)} \tag{16}$$

Here, $B$ is the batch size, $\text{sim}(\cdot, \cdot)$ denotes similarity (dot product) and $\tau$ is a tunable temperature.

Let $h^{a2}$ denote the concatenated representation across scales for the second augmentation. To enhance augmentation learning, we introduce two adversarial pairwise contrastive losses:

$$\mathcal{L}^{\text{pair1}} = \text{sim}(\mathbf{h}_i^a, \mathbf{h}_i^{a1}) - \text{sim}(\mathbf{h}_i, \mathbf{h}_i^a)$$
$$\mathcal{L}^{\text{pair2}} = \text{sim}(\mathbf{h}_i, \mathbf{h}_{i'}) - \text{sim}(\mathbf{h}_i^a, \mathbf{h}_i^{a1}) \tag{17}$$

where $\mathbf{h}_{i'}$ is a randomly shuffled sample from the same batch. The first loss enforces fidelity by keeping each augmentation $\mathbf{h}_i^a$ closer to its original $\mathbf{h}_i$ than to another augmentation $\mathbf{h}_i^{a1}$, while the second enforces invariance by pulling augmentations of the same sample closer than unrelated ones. This adversarial design strengthens augmentation learning.

The total loss is a weighted sum of predictive and contrastive terms.

$$\mathcal{L}^{\text{final}} = \mathcal{L} + \beta 1 \mathcal{L}^{\text{auxiliary}} + \beta_2 \mathcal{L}^{\text{pair1}} + \beta_3 \mathcal{L}^{\text{pair2}} \tag{18}$$

where $\beta_1$, $\beta_2$, and $\beta_3$ are hyperparameters weighting the auxiliary terms.

## 4 EXPERIMENTS

### 4.1 EXPERIMENTAL SETTINGS

#### 4.1.1 BENCHMARKS

We evaluate FreqMixAttNet on six popular datasets: ETTh1, ETTh2, ETTm1, ETTm2, Weather, and Exchange-rate. We also measure dataset forecastability Goerg (2013), noting that ETTh*,

Weather, and Exchange-rate have relatively low values, making them challenging benchmarks. Dataset statistics are shown in Table 1, with more details in the Appendix A.1.

Table 1: Details of benchmark datasets. Higher forecastability indicates better predictability.

| Datasets | ETTh1 | ETTh2 | ETTm1 | ETTm2 | Weather | Exchange_Rate |
|---|---|---|---|---|---|---|
| Variate | 7 | 7 | 7 | 7 | 21 | 8 |
| Timesteps | 17420 | 17420 | 69680 | 69680 | 52696 | 7587 |
| Forecastability | 0.38 | 0.45 | 0.46 | 0.55 | 0.33 | 0.41 |

### 4.1.2 BASELINES AND SETUPS

We evaluate our method against a wide range of SOTA architectures. Since baseline results obtained with different input lengths and hyperparameter search strategies are not directly comparable, we fix the input length to $L = 96$ for all prediction horizons $T \in 96, 192, 336, 720$ and include only baselines with input length 96 in the experimental results. The baseline results are collected from Chen et al. (2025) or the original papers.

Specifically, in the time domain, we include CNN-based models (MICN Wang et al. (2023), Times-Net Wu et al. (2023)), an MLP-based model (DLinear Zeng et al. (2023)), and Transformer-based models (iTransformer Liu et al. (2024), PatchTST Nie et al. (2023b)) as baselines. For multi-scale trend–season decomposition, we adopt TimeMixer Wang et al. (2024a). We consider two strong baselines, CATS Lu et al. (2024) (time domain) and SimpleTM Chen et al. (2025) (frequency domain), which employ complex Transformer architectures. In addition, we adopt ATFNet Ye & Gao (2024) as a baseline, which models the time and frequency domains separately without interaction. Performance is evaluated using mean squared error (MSE) and mean absolute error (MAE) on multivariate time series forecasting.

## 4.2 MAIN RESULTS

### 4.2.1 OVERALL PERFORMANCE.

To facilitate comparison, we evaluate multiple forecast horizons with a fixed input sequence length of 96 for long-term forecasting. Detailed settings are provided in Appendix A.1. Table 2 summarizes the forecasting performance across all datasets and baselines. Compared with strong baselines in the frequency domain (SimpleTM Chen et al. (2025)) and the time domain (CATS Lu et al. (2024)), both of which employ complex Transformer architectures, our proposed model FreqMixAttention achieves superior performance. Compared with TimeMixer Wang et al. (2024a) and ATFNet Ye & Gao (2024), our model delivers better results, suggesting that multi-scale trend–season decomposition alone or modeling the time and frequency domains separately without interaction is insufficient. FreqMixAttention attains the best results on most forecast horizons (17/24 for MSE and 20/24 for MAE). These findings highlight that integrating both domains is essential for capturing underlying patterns in diverse time series data.

### 4.2.2 ADDITIONAL ANALYSIS

**Ablation Study**

We further conducted an ablation study to evaluate four key architectural components: Adaptive Convolution Embedding (ACE), Wavelet Decomposition (WD), Dual-domain Cross Attention (DCA), and the Auxiliary Contrastive Loss (ACL). Table 3 summarizes the results on four ETT datasets, where each component is removed individually under the 96-step prediction horizon. Removing ACE and DCA leads to significant performance degradation, while WD provides slight improvements. The ACL also contributes marginally more than WD.

**Contrastive Auxiliary Strategies**

To further examine the design choices within the Contrastive Auxiliary Loss, we developed five variants (Cases 2–6) focusing on three key factors: augmentation type, output layer selection, and output layer aggregation. Detailed results and additional ablation studies are provided in Appendix A.5. Our main findings are as follows: (1) The choice of output layer, aggregation method, and

Table 2: Multivariate long-term forecasting results with a fixed lookback window (L = 96). Results are reported for four prediction lengths: 96, 192, 336, 720. Lower MSE/MAE indicates better performance. Best results are in bold, second-best are underlined.

| Models | | FreqMix-AttNet(ours) | | SimpleTM (2025) | | ATFNet (2024) | | CATS (2024) | | TimeMixer (2024) | | iTransformer (2024) | | PatchTST (2023) | | TimesNet (2023) | | MICN (2023) | | Dlinear (2023) | |
|---|---|---|---|---|---|---|---|---|---|---|---|---|---|---|---|---|---|---|---|---|---|
| Metric | | MSE | MAE | MSE | MAE | MSE | MAE | MSE | MAE | MSE | MAE | MSE | MAE | MSE | MAE | MSE | MAE | MSE | MAE | MSE | MAE |
| ETTh1 | 96 | **0.362** | **0.387** | 0.366 | 0.392 | 0.413 | 0.441 | 0.371 | 0.395 | 0.375 | 0.400 | 0.386 | 0.405 | 0.460 | 0.447 | 0.384 | 0.402 | 0.426 | 0.446 | 0.397 | 0.412 |
| | 192 | **0.414** | **0.417** | 0.422 | 0.421 | 0.468 | 0.483 | 0.426 | 0.422 | 0.429 | 0.421 | 0.441 | 0.436 | 0.512 | 0.477 | 0.436 | 0.429 | 0.454 | 0.464 | 0.446 | 0.441 |
| | 336 | 0.438 | 0.433 | 0.440 | 0.438 | 0.551 | 0.536 | **0.437** | **0.432** | 0.501 | 0.462 | 0.484 | 0.458 | 0.546 | 0.496 | 0.638 | 0.469 | 0.493 | 0.487 | 0.489 | 0.467 |
| | 720 | **0.451** | **0.461** | 0.463 | 0.462 | 0.666 | 0.604 | 0.474 | **0.461** | 0.498 | 0.482 | 0.503 | 0.491 | 0.544 | 0.517 | 0.521 | 0.500 | 0.526 | 0.526 | 0.513 | 0.510 |
| ETTh2 | 96 | **0.276** | **0.328** | 0.281 | 0.338 | 0.295 | 0.344 | 0.287 | 0.341 | 0.289 | 0.341 | 0.297 | 0.349 | 0.308 | 0.355 | 0.340 | 0.374 | 0.372 | 0.424 | 0.340 | 0.394 |
| | 192 | **0.350** | **0.374** | 0.355 | 0.387 | 0.390 | 0.408 | 0.361 | 0.388 | 0.372 | 0.392 | 0.380 | 0.400 | 0.393 | 0.405 | 0.402 | 0.414 | 0.492 | 0.492 | 0.482 | 0.479 |
| | 336 | 0.392 | 0.410 | **0.365** | **0.401** | 0.465 | 0.464 | 0.374 | 0.403 | 0.386 | 0.414 | 0.428 | 0.432 | 0.427 | 0.436 | 0.452 | 0.452 | 0.607 | 0.555 | 0.591 | 0.541 |
| | 720 | **0.411** | **0.433** | 0.413 | 0.436 | 0.515 | 0.513 | 0.412 | 0.433 | 0.412 | 0.434 | 0.427 | 0.445 | 0.436 | 0.450 | 0.462 | 0.468 | 0.824 | 0.655 | 0.839 | 0.661 |
| ETTm1 | 96 | **0.310** | **0.347** | 0.321 | 0.361 | 0.339 | 0.375 | 0.318 | 0.361 | 0.320 | 0.357 | 0.334 | 0.368 | 0.352 | 0.374 | 0.338 | 0.375 | 0.365 | 0.387 | 0.346 | 0.374 |
| | 192 | **0.356** | **0.376** | 0.360 | 0.380 | 0.367 | 0.388 | 0.357 | 0.377 | 0.361 | 0.381 | 0.377 | 0.391 | 0.390 | 0.393 | 0.374 | 0.387 | 0.403 | 0.408 | 0.382 | 0.391 |
| | 336 | **0.384** | **0.397** | 0.390 | 0.404 | 0.392 | 0.406 | 0.387 | 0.401 | 0.390 | 0.404 | 0.426 | 0.420 | 0.421 | 0.414 | 0.410 | 0.411 | 0.436 | 0.431 | 0.415 | 0.415 |
| | 720 | **0.443** | **0.432** | 0.454 | 0.438 | 0.453 | 0.444 | 0.448 | 0.437 | 0.458 | 0.441 | 0.491 | 0.459 | 0.462 | 0.449 | 0.478 | 0.450 | 0.489 | 0.462 | 0.473 | 0.451 |
| ETTm2 | 96 | **0.169** | **0.249** | 0.173 | 0.257 | 0.178 | 0.266 | 0.178 | 0.261 | 0.175 | 0.258 | 0.180 | 0.264 | 0.183 | 0.270 | 0.187 | 0.267 | 0.197 | 0.296 | 0.193 | 0.293 |
| | 192 | **0.233** | **0.294** | 0.238 | 0.299 | 0.260 | 0.324 | 0.248 | 0.308 | 0.237 | 0.299 | 0.250 | 0.309 | 0.255 | 0.314 | 0.249 | 0.309 | 0.284 | 0.361 | 0.284 | 0.361 |
| | 336 | **0.291** | **0.331** | 0.296 | 0.338 | 0.328 | 0.363 | 0.304 | 0.343 | 0.298 | 0.340 | 0.311 | 0.348 | 0.309 | 0.347 | 0.321 | 0.351 | 0.381 | 0.429 | 0.382 | 0.429 |
| | 720 | **0.388** | **0.389** | 0.275 | 0.322 | 0.448 | 0.435 | 0.402 | 0.402 | 0.391 | 0.396 | 0.412 | 0.407 | 0.412 | 0.404 | 0.408 | 0.403 | 0.549 | 0.522 | 0.558 | 0.525 |
| Weather | 96 | 0.162 | **0.203** | 0.162 | 0.207 | 0.173 | 0.221 | **0.161** | 0.207 | 0.163 | 0.209 | 0.174 | 0.214 | 0.186 | 0.227 | 0.172 | 0.220 | 0.198 | 0.261 | 0.195 | 0.252 |
| | 192 | **0.208** | **0.246** | **0.208** | 0.248 | 0.214 | 0.259 | **0.208** | 250 | **0.208** | 0.250 | 0.221 | 0.254 | 0.234 | 0.265 | 0.219 | 0.261 | 0.239 | 0.299 | 0.237 | 0.295 |
| | 336 | 0.265 | 0.288 | 0.263 | 0.290 | 0.264 | 0.297 | 0.264 | 0.290 | 0.251 | **0.287** | 0.278 | 0.296 | 0.284 | 0.301 | **0.246** | 0.337 | 0.285 | 0.336 | 0.282 | 0.331 |
| | 720 | 0.344 | **0.340** | 0.340 | 0.341 | 0.332 | 0.343 | 0.342 | 0.341 | **0.339** | 0.341 | 0.358 | 0.347 | 0.356 | 0.349 | 0.365 | 0.359 | 0.351 | 0.388 | 0.345 | 0.382 |
| Exchange-Rate | 96 | 0.088 | **0.205** | 0.094 | 0.215 | 0.095 | 0.218 | **0.085** | **0.205** | 0.090 | 0.235 | 0.086 | 0.206 | 0.088 | **0.205** | 0.107 | 0.234 | 0.093 | 0.228 | 0.088 | 0.218 |
| | 192 | **0.176** | **0.299** | 0.177 | 0.301 | 0.228 | 0.339 | 0.177 | **0.299** | 0.187 | 0.343 | 0.177 | **0.299** | **0.176** | **0.299** | 0.226 | 0.344 | 0.185 | 0.332 | **0.176** | 0.315 |
| | 336 | **0.301** | **0.397** | 0.379 | 0.449 | 0.342 | 0.431 | 0.326 | 0.412 | 0.353 | 0.473 | 0.331 | 0.417 | **0.301** | **0.397** | 0.367 | 0.448 | 0.354 | 0.468 | 0.313 | 0.427 |
| | 720 | 0.917 | 0.720 | 0.924 | 0.726 | 0.848 | 0.712 | **0.837** | 0.690 | 0.934 | 0.761 | 0.847 | 0.691 | 0.901 | 0.714 | 0.964 | 0.746 | 0.872 | 0.727 | 0.839 | 0.695 |
| $1^{st}$ Count | | **17** | **20** | 2 | 1 | 0 | 0 | 5 | 6 | 2 | 1 | 0 | 1 | 2 | 3 | 1 | 0 | 0 | 0 | 1 | 0 |

Table 3: Ablation Study on ACE, WD, DCA, and ACL for Long-Term Forecasting on Four ETT Datasets. '1' denotes the default design in FreqMixAttNet . Best results are highlighted in bold.

| Case | Adaptive convolution (ACE) | Wavelet Decomposition (WD) | dual-domain cross-attention (DCA) | Contrastive auxiliary (ACL) | ETTh1 | | ETTh2 | | ETTm1 | | ETTm2 | |
|---|---|---|---|---|---|---|---|---|---|---|---|---|
| | | | | | MSE | MAE | MSE | MAE | MSE | MAE | MSE | MAE |
| 1 | ✓ | ✓ | ✓ | ✓ | **0.362** | **0.387** | **0.276** | **0.328** | **0.310** | **0.347** | **0.169** | **0.249** |
| 2 | ✗ | ✓ | ✓ | ✓ | 0.380 | 0.392 | 0.352 | 0.387 | 0.419 | 0.355 | 0.178 | 0.257 |
| 3 | ✓ | ✗ | ✓ | ✓ | 0.363 | 0.389 | 0.280 | 0.331 | 0.311 | 0.349 | 0.173 | 0.254 |
| 4 | ✓ | ✓ | ✗ | ✓ | 0.378 | 0.388 | 0.288 | 0.335 | 0.343 | 0.363 | 0.182 | 0.259 |
| 5 | ✓ | ✓ | ✓ | ✗ | 0.366 | 0.392 | 0.297 | 0.331 | 0.312 | 0.350 | 0.172 | 0.252 |

augmentation type all influence model performance. (2) These design choices have a stronger impact on fine-grained datasets (ETTm*) than on coarse-grained datasets (ETTh*).

### 4.2.3 KEY PARAMETERS ANALYSIS

In addition, we conducted hyperparameter sensitivity analyses on the learning rate, the dimension of the adaptive convolution module (d_model), the number of cross attention heads (n_heads), and the augmentation weight(augment_weight), as shown in Figure 2 on the ETTh1 dataset. More details

Table 4: Ablation of Contrastive Auxiliary Strategies on Four ETT Datasets (Forecast Length = 96). 1 is the official design in FreqMixAttNet .

| Case | Contrastive output layer | | | Augmentation type | | | Aggregation method | | ETTh1 | | ETTh2 | | ETTm1 | | ETTm2 | |
|------|-----------|---------|------------|------|------|-----------|-----------|--------|-----|-----|-----|-----|-----|-----|-----|-----|
| | predicter | encoder | projection | freq | time | mixing-up | stack&sum | concat | MSE | MAE | MSE | MAE | MSE | MAE | MSE | MAE |
| 1 | ✓ | × | × | ✓ | × | × | ✓ | × | 0.362 | 0.387 | 0.276 | 0.328 | 0.310 | 0.347 | 0.169 | 0.249 |
| 2 | × | ✓ | × | ✓ | × | × | ✓ | × | 0.363 | 0.388 | 0.285 | 0.331 | 0.312 | 0.349 | 0.172 | 0.252 |
| 3 | × | × | ✓ | ✓ | × | × | ✓ | × | 0.365 | 0.392 | 0.279 | 0.331 | 0.312 | 0.350 | 0.174 | 0.254 |
| 4 | ✓ | × | × | × | ✓ | × | ✓ | × | 0.362 | 0.387 | 0.278 | 0.329 | 0.313 | 0.351 | 0.174 | 0.254 |
| 5 | ✓ | × | × | × | × | ✓ | ✓ | × | 0.362 | 0.388 | 0.278 | 0.329 | 0.313 | 0.350 | 0.173 | 0.253 |
| 6 | ✓ | × | × | ✓ | × | × | × | ✓ | 0.364 | 0.387 | 0.280 | 0.330 | 0.310 | 0.348 | 0.176 | 0.256 |

can be found in the Appendix A.6. Specifically, we find that the model is sensitive to the learning

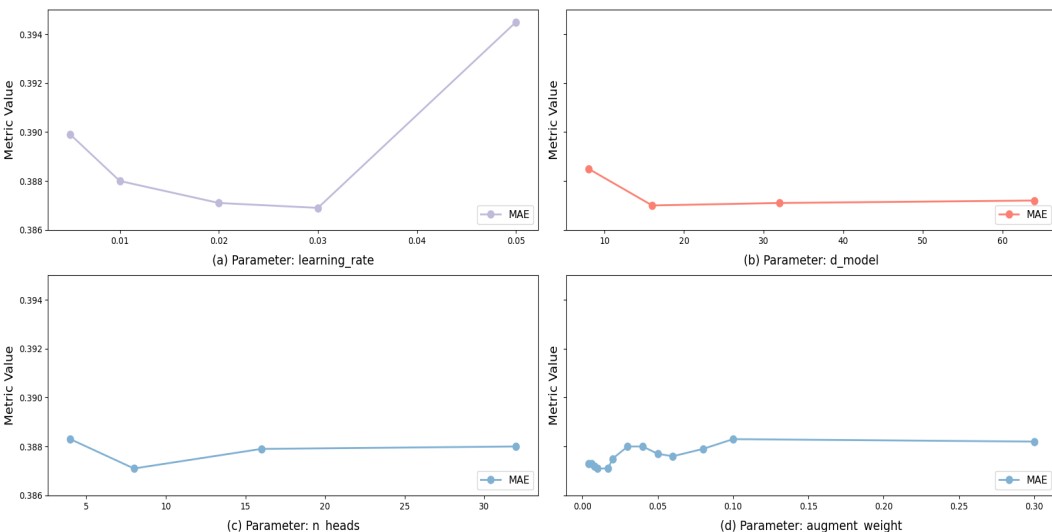

Figure 2: Hyperparameter Sensitivity: learning_rate, d_model, n_heads, and augment_weight

rate, indicating that proper hyperparameter tuning is necessary for each model. Both d_model and n_heads significantly affect performance, with 16 and 8 yielding the best results on the ETTh1 dataset, respectively. In addition, the augment_weight should also be carefully tuned.

## 5 CONCLUSION

We propose FreqMixAttNet, a unified cross-domain framework that fuses time- and frequency-domain representations. Through adaptive decomposition, cross-domain attention, and contrastive regularization, it achieves state-of-the-art forecasting performance across benchmarks.

## 6 ETHICS STATEMENT

Our work only focuses on the scientific problem, so there is no potential ethical risk.We adopt an LLM to improve grammatical correctness.

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

# A APPENDIX

## A.1 IMPLEMENTATION DETAILS

We summarized details of datasets, evaluation metrics, baseline models in this section.

**DATASETS** We evaluate the performance of different models for long-term forecasting on 6 well-established datasets, including ETT (Electricity Transformer Temperature) datasets (ETTh1, ETTh2, ETTm1, ETTm2), Weather and Exchange-rate. ETT[1] datasets are collected from two different electric transformers labeled with 1 and 2, and each of them contains 2 different resolutions (15 minutes and 1 hour) denoted with m and h. Weather[2] dataset collects 21 meteorological indicators in Germany, such as humidity and air temperature. c[3] dataset collects 137 photovoltaic (PV) plants in the United States, including data on size, location, cost, and PV technology used. Exchange_Rate[4] collects the panel data of daily exchange rates from 8 countries from 1990 to 2016. Furthermore, We detail the descriptions of the dataset in Table 5.

---

[1]https://github.com/zhouhaoyi/ETDataset

[2]https://www.bgc-jena.mpg.de/wetter

[3]https://www.nrel.gov/grid/solar-power-data.html

[4]https://github.com/laiguokun/multivariate-time-series-data

Table 5: Details of 6 real-world datasets. The dataset size is organized in (Train, Validation, Test).

| Tasks | Dataset | Dimension | lookback window | Forecasting Horizon | Frequency | Dataset Size | Description |
|---|---|---|---|---|---|---|---|
| Long-term Forecasting | ETTh1 | 7 | 96 | {96, 192, 336, 720} | 1hour | (8545, 2881, 2881) | Temperature |
| | ETTh2 | 7 | 96 | {96, 192, 336, 720} | 1hour | (8545, 2881, 2881) | Temperature |
| | ETTm1 | 7 | 96 | {96, 192, 336, 720} | 15min | (34465, 11521, 11521) | Temperature |
| | ETTm2 | 7 | 96 | {96, 192, 336, 720} | 15min | (34465, 11521, 11521) | Temperature |
| | Weather | 21 | 96 | {96, 192, 336, 720} | 10min | (36792, 5271, 10540) | Weather |
| | Exchange-rate | 8 | 96 | {96, 192, 336, 720} | Daily | (5120, 665, 1422) | Weather |

**METRICS** We calculate the Mean Squared Error (MSE) and Mean Absolute Error (MAE) of multivariate time series forecasting as metrics, which is widely used in Time Series forecasting. The calculations of these metrics are:

$$MSE = \sum_{i=1}^{P} (\mathbf{X}_i - \hat{\mathbf{X}}_i)^2$$
$$MAE = \sum_{i=1}^{P} |\mathbf{X}_i - \hat{\mathbf{X}}_i| \tag{19}$$

where $\mathbf{X}, \hat{\mathbf{X}} \in \mathbb{R}^{P*C}$ are the ground truth and prediction results of the future with $P$ time slices and $C$ dimensions, $\mathbf{X}_i$ means the i-th future time point.

**BASELINES** Time series analysis has a long history, and with the rise of deep learning in long-term forecasting, we select nine advanced baseline models spanning diverse architectures to evaluate the effectiveness of our method.

We first adopt two CNN-based models, MICN Wang et al. (2023) and TimesNet Wu et al. (2023). MICN employs multi-scale 1D and 2D convolutions to model time series, while TimesNet reshapes time series into 2D tensors based on different periods and then applies 2D convolutions, both introducing complex convolutional architectures.

Next, we adopt an MLP-based model, DLinear Zeng et al. (2023), a strong baseline that uses a MLP for time series modeling. For Transformer-based baselines, we include iTransformer Liu et al. (2024) and PatchTST Nie et al. (2023b). iTransformer applies the Transformer along the feature dimension to enhance feature interactions, while PatchTST operates on patches to capture local temporal dependencies.

For multi-scale trend–season decomposition, we adopt TimeMixer Wang et al. (2024a). We also consider two strong Transformer-based baselines: CATS Lu et al. (2024), which constructs auxiliary time series via a 2D temporal-contextual attention mechanism to capture inter-series relationships efficiently, and SimpleTM Chen et al. (2025), a lightweight model that uses signal processing–based tokenization and geometric algebra–enhanced attention, achieving competitive results with much larger Transformer architectures.

Finally, to assess the effectiveness of time–frequency domain fusion, we adopt ATFNet Ye & Gao (2024), which models the two domains separately without interaction.

**EXPERIMENT DETAILS** All experiments were implemented in Pytorch, and conducted on multi NVIDIA H20 100GB GPUs and CUDA Version is 12.6. We used the ADAM optimizer with L2 loss for model optimization.

For baseline comparisons, input length and hyperparameter tuning strategies can substantially affect performance. To ensure fairness, we consistently report results with an input length of $L = 96$. Whenever possible, we directly use results reported in the original papers or in Chen et al. (2025), since the original authors are in the best position to conduct hyperparameter searches for their models. For ATFNet Ye & Gao (2024), results with $L = 96$ were not available; hence, we fine-tuned the model with hyperparameter search based on the official code.For baselines without reported results in the Exchange_rate dataset, we carefully reproduced them following the released implementations.

**HYPERPARAMETER SETTINGS** Our hyperparameter selection followed a systematic approach, combining grid search with domain-specific considerations. The input length L was set to 96 to ensure fair comparisons across benchmark datasets. For training parameters of each dataset, we performed a grid search over learning rates within a logarithmic scale from $1 * 10^{-4}$ to $5 * 10^{-2}$, the number of layers was systematically evaluated within the ranges $\{1, 2, 3, 4\}$, the batch size set $\{16, 32, 64, 128\}$, the dropout set 0 to $2 * 10^{-1}$, the number of multi-heads set $\{4, 8\}$ and the d_model of embedding set $\{8, 16, 32\}$.

A.2   DATA AUGMENTATIONS

A critical aspect of contrastive learning involves the selection of appropriate augmentation strategies that introduce meaningful priors for constructing feasible positive samples, thereby enabling encoders to learn robust and discriminative representations. To evaluate the effectiveness of different augmentation approaches, we compare three methods: time-domain augmentation, frequency-domain augmentation, and a mixed augmentation. These are rigorously examined through a series of complementary experiments.

**TIME-DOMAIN AUGMENTATION** We employ jittering augmentation (Um et al. (2017)), a technique that introduces random noise sampled from a Gaussian distribution $\mathcal{N}(0; 0.3)$ to the input time series. This method enhances model robustness against both multiplicative and additive noise disturbances and contributes to improved generalization performance.

**FREQUENCY-DOMAIN AUGMENTATION** Inspired by time-domain jitter augmentation, we designed a analogous structure tailored for the frequency domain. Unlike its time-domain counterpart, however, the proposed method incorporates frequency-domain operations motivated by frequency masking techniques in signal processing. By artificially removing or adding frequency components, that is:

- Frequency Component Removal: By applying a random mask to zero out a subset of frequency components, this operation simulates frequency attenuation during signal transmission or incomplete sensor capture. This forces the model to avoid over-reliance on specific frequency bands and enhances its ability to identify critical features.
- Limited Noise Injection: This method adds small amplitude random perturbations constrained to specific frequency bands, mimicking environmental noise or device interference. Such controlled perturbation preserves the semantic structure of the original signal while improving the diversity of the data.

Below is the pseudocode for the frequency-domain data augmentation algorithm:

---

**Algorithm 1** Frequency-Domain Data Augmentation

---

**Input:** Original time series $\boldsymbol{x}$, mask rate of frequency $r$, amplitude limitation $\epsilon$
**Output:** Augmentation time series $\boldsymbol{o}$
1  $mask_1, mask_2 \leftarrow$ uniform random matrix with shape equal to $\boldsymbol{x}$, each element $\sim \mathcal{U}(0; 1)$
2  $mask_1 \leftarrow mask_1 > r, mask_2 \leftarrow mask_2 > (1 - r)$
3  $aug_1 \leftarrow mask_1 \odot \boldsymbol{x}$
4  $R_{am} \leftarrow$ gaussian matrix with shape equal to $mask_2$, each element $\sim \mathcal{N}(0; 1)$
5  $m_a \leftarrow Max(|\boldsymbol{x}|) * \epsilon$ {Compute maximum amplitude of the input within amplitude limitation}
6  $R_{am} \leftarrow R_{am} \times m_a$
7  $aug_2 \leftarrow mask_2 \odot R_{am}$
8  $\boldsymbol{o} \leftarrow aug_1 + aug_2$
9  **return** $\boldsymbol{o}$

---

This approach simulates partial signal loss or noise interference in real-world scenarios, thereby compelling the model to learn more robust feature representations.

**MIXING-UP AUGMENTATION** Frequency mixing is employed to generate a novel contextual view by substituting a specific proportion of frequency components derived through Fast Fourier Transform (FFT) from one training instance with the corresponding frequency components of another randomly selected training instance within the same batch. The modified frequency-domain

representation is subsequently transformed back into the time domain via inverse FFT, resulting in a synthetic time series (Chen et al., 2023). This method of intersample frequency exchange avoids introducing extraneous noise or artificial periodicities, thereby providing more reliable augmentations that better preserve the semantic integrity and intrinsic characteristics of the original data.

## A.3 CONTRASTIVE AUXILIARY LOSS

Following the methodology of Liu & Chen (2024), we adopted a contrastive loss that integrates both temporal and sample dimensions, which is the auxiliary loss mentioned in section 3.2.5. The impact of the key parameter alpha in this framework is further analyzed in the appendix A.6.

## A.4 FULL RESULTS

In the long-term forecasting results presented in Table 2 of the main paper, we reported four prediction lengths. Table 6 provides a comprehensive breakdown of empirical results for each prediction length and the averaged performance. Within each row, the lowest MSE and MAE scores are highlighted in **red**, and the second-lowest scores are underscored in blue. Our proposed method consistently achieves near top-2 performance across all evaluations. To ensure a fair comparison between models, we cite the baseline results mainly from SimpleTM (2025) and CATS(2024). We can find that the relative promotion of FreqMixAttNet is a practical model in real-world applications and is valuable to deep time series forecasting community.

## A.5 ADDITIONAL ABLATION STUDIES

Here we provide the complete results of ablations and alternative designs for FreqMixAttNet .

### A.5.1 ABLATIONS ON ARCHITECTURAL COMPONENTS

To rigorously validate our approach, we conducted additional experiments across four datasets (ETTh1, ETTh2, ETTm1, ETTm2) under fixed 96-step forecasting settings. The comprehensive results in Table 7 reveal three critical findings: (1) The complete architecture achieves optimal performance, (2) Key innovations demonstrate non-interchangeable advantages over traditional methods, and (3) Different components exhibit dataset-agnostic improvement patterns.

**Implementations** We designed six ablation scenarios through progressive component replacement:

- Offical design in FreqMixAttNet (case 1).

- Ablations on Wavelet Decomposition (case 2): In this case, we replace the frequency-domain wavelet time-series decomposition with the moving-average-based season-trend decomposition of time-domain, which is widely used in previous work, such as Autoformer (Wu et al. (2021)), FEDformer (Zhou et al. (2022)).

- Ablations on Adaptive Convolution of embedding (case 3): In this case, We only adopt a standard 1D convolution rather than an adaptive convolution.

- Ablations on Dual-Domain Cross-Attention (case 4): In this case, We remove the mixing operation of frequency-domain and time-domain into a cross attention in seasonal part.

- Ablations on contrastive Loss (case 5-6): Firstly, We remove the contrastive loss between the base sequence representation and the augmented sequence representation. Then, We removed the pairwise loss between two pairs of ordered representations, namely the adversarial learning between different augmented representations.

**Analysis** In all ablations, we can find that the official design in FreqMixAttNet performs best, which provides solid support to our insights in special architecture. Notably, it is observed that completely reversing adaptive convolution, dual-domain cross-attention and contrastive auxiliary loss (case 3-6) leads to a seriously peformance drop. This may come from that:

- Adaptive Convolution enhances the input-adaptive learning capability of representations across all time series.

Table 6: Multivariate long-term forecasting results with a unified lookback window L=96. Full results of 4 different prediction lengths, that is 96, 192, 336, 720. A lower MSE or MAE indicates a better prediction. The best results are in bold and the second best are underlined.

| Models | | FreqMix-AttNet(ours) | | SimpleTM (2025) | | ATFNet (2024) | | CATS (2024) | | TimeMixer (2024) | | iTransformer (2024) | | PatchTST (2023) | | TimesNet (2023) | | MICN (2023) | | Dlinear (2023) | |
|---|---|---|---|---|---|---|---|---|---|---|---|---|---|---|---|---|---|---|---|---|---|
| Metric | | MSE | MAE | MSE | MAE | MSE | MAE | MSE | MAE | MSE | MAE | MSE | MAE | MSE | MAE | MSE | MAE | MSE | MAE | MSE | MAE |
| ETTh1 | 96 | **0.362** | **0.387** | 0.366 | 0.392 | 0.413 | 0.441 | 0.371 | 0.395 | 0.375 | 0.400 | 0.386 | 0.405 | 0.460 | 0.447 | 0.384 | 0.402 | 0.426 | 0.446 | 0.397 | 0.412 |
| | 192 | **0.414** | **0.417** | 0.422 | 0.421 | 0.468 | 0.483 | 0.426 | 0.422 | 0.429 | 0.421 | 0.441 | 0.436 | 0.512 | 0.477 | 0.436 | 0.429 | 0.454 | 0.464 | 0.446 | 0.441 |
| | 336 | 0.438 | 0.433 | 0.440 | 0.438 | 0.551 | 0.536 | **0.437** | **0.432** | 0.501 | 0.462 | 0.484 | 0.458 | 0.546 | 0.496 | 0.638 | 0.469 | 0.493 | 0.487 | 0.489 | 0.467 |
| | 720 | **0.451** | **0.461** | 0.463 | 0.462 | 0.666 | 0.604 | 0.474 | **0.461** | 0.498 | 0.482 | 0.503 | 0.491 | 0.544 | 0.517 | 0.521 | 0.500 | 0.526 | 0.526 | 0.513 | 0.510 |
| | avg | **0.416** | **0.425** | 0.423 | 0.428 | 0.525 | 0.516 | 0.427 | 0.428 | 0.447 | 0.440 | 0.454 | 0.448 | 0.516 | 0.484 | 0.495 | 0.450 | 0.475 | 0.481 | 0.461 | 0.458 |
| ETTh2 | 96 | **0.276** | **0.328** | 0.281 | 0.338 | 0.295 | 0.344 | 0.287 | 0.341 | 0.289 | 0.341 | 0.297 | 0.349 | 0.308 | 0.355 | 0.340 | 0.374 | 0.372 | 0.424 | 0.340 | 0.394 |
| | 192 | **0.350** | **0.374** | 0.355 | 0.387 | 0.390 | 0.408 | 0.361 | 0.388 | 0.372 | 0.392 | 0.380 | 0.400 | 0.393 | 0.405 | 0.402 | 0.414 | 0.492 | 0.492 | 0.482 | 0.479 |
| | 336 | 0.392 | 0.410 | **0.365** | **0.401** | 0.465 | 0.464 | 0.374 | 0.403 | 0.386 | 0.414 | 0.428 | 0.432 | 0.427 | 0.436 | 0.452 | 0.452 | 0.607 | 0.555 | 0.591 | 0.541 |
| | 720 | **0.411** | **0.433** | 0.413 | 0.436 | 0.515 | 0.513 | 0.412 | 0.433 | 0.412 | 0.434 | 0.427 | 0.445 | 0.436 | 0.450 | 0.462 | 0.468 | 0.824 | 0.655 | 0.839 | 0.661 |
| | avg | 0.411 | 0.433 | **0.354** | **0.391** | 0.416 | 0.432 | 0.359 | **0.391** | 0.364 | 0.395 | 0.383 | 0.407 | 0.391 | 0.412 | 0.414 | 0.427 | 0.574 | 0.532 | 0.563 | 0.519 |
| ETTm1 | 96 | **0.310** | **0.347** | 0.321 | 0.361 | 0.339 | 0.375 | 0.318 | 0.361 | 0.320 | 0.357 | 0.334 | 0.368 | 0.352 | 0.374 | 0.338 | 0.375 | 0.365 | 0.387 | 0.346 | 0.374 |
| | 192 | **0.356** | **0.376** | 0.360 | 0.380 | 0.367 | 0.388 | 0.357 | 0.377 | 0.361 | 0.381 | 0.377 | 0.391 | 0.390 | 0.393 | 0.374 | 0.387 | 0.403 | 0.408 | 0.382 | 0.391 |
| | 336 | **0.384** | **0.397** | 0.390 | 0.404 | 0.392 | 0.406 | 0.387 | 0.401 | 0.390 | 0.404 | 0.426 | 0.420 | 0.421 | 0.414 | 0.410 | 0.411 | 0.436 | 0.431 | 0.415 | 0.415 |
| | 720 | **0.443** | **0.432** | 0.454 | 0.438 | 0.453 | 0.444 | 0.448 | 0.437 | 0.458 | 0.441 | 0.491 | 0.459 | 0.462 | 0.449 | 0.478 | 0.450 | 0.489 | 0.462 | 0.473 | 0.451 |
| | avg | **0.373** | **0.388** | 0.381 | 0.396 | 0.388 | 0.403 | 0.378 | 0.393 | 0.381 | 0.395 | 0.407 | 0.410 | 0.406 | 0.408 | 0.400 | 0.406 | 0.423 | 0.422 | 0.404 | 0.408 |
| ETTm2 | 96 | **0.169** | **0.249** | 0.173 | 0.257 | 0.178 | 0.266 | 0.178 | 0.261 | 0.175 | 0.258 | 0.180 | 0.264 | 0.183 | 0.270 | 0.187 | 0.267 | 0.197 | 0.296 | 0.193 | 0.293 |
| | 192 | **0.233** | **0.294** | 0.238 | 0.299 | 0.260 | 0.324 | 0.248 | 0.308 | 0.237 | 0.299 | 0.250 | 0.309 | 0.255 | 0.314 | 0.249 | 0.309 | 0.284 | 0.361 | 0.284 | 0.361 |
| | 336 | **0.291** | **0.331** | 0.296 | 0.338 | 0.328 | 0.363 | 0.304 | 0.343 | 0.298 | 0.340 | 0.311 | 0.348 | 0.309 | 0.347 | 0.321 | 0.351 | 0.381 | 0.429 | 0.382 | 0.429 |
| | 720 | 0.388 | 0.389 | 0.275 | 0.322 | 0.448 | 0.435 | 0.402 | 0.402 | 0.391 | 0.396 | 0.412 | 0.407 | 0.412 | 0.404 | 0.408 | 0.403 | 0.549 | 0.522 | 0.558 | 0.525 |
| | avg | **0.270** | **0.316** | 0.275 | 0.322 | 0.303 | 0.347 | 0.283 | 0.329 | 0.275 | 0.323 | 0.288 | 0.332 | 0.290 | 0.334 | 0.291 | 0.333 | 0.353 | 0.402 | 0.354 | 0.402 |
| Weather | 96 | 0.162 | **0.203** | 0.162 | 0.207 | 0.173 | 0.221 | **0.161** | 0.207 | 0.163 | 0.209 | 0.174 | 0.214 | 0.186 | 0.227 | 0.172 | 0.220 | 0.198 | 0.261 | 0.195 | 0.252 |
| | 192 | **0.208** | **0.246** | **0.208** | 0.248 | 0.214 | 0.259 | **0.208** | 250 | **0.208** | 0.250 | 0.221 | 0.254 | 0.234 | 0.265 | 0.219 | 0.261 | 0.239 | 0.299 | 0.237 | 0.295 |
| | 336 | 0.265 | 0.288 | 0.263 | 0.290 | 0.264 | 0.297 | 0.264 | 0.290 | 0.251 | **0.287** | 0.278 | 0.296 | 0.284 | 0.301 | **0.246** | 0.337 | 0.285 | 0.336 | 0.282 | 0.331 |
| | 720 | 0.344 | **0.340** | 0.340 | 0.341 | 0.332 | 0.343 | 0.342 | 0.341 | **0.339** | 0.341 | 0.358 | 0.347 | 0.356 | 0.349 | 0.365 | 0.359 | 0.351 | 0.388 | 0.345 | 0.382 |
| | avg | 0.245 | **0.269** | 0.243 | 0.272 | 0.246 | 0.280 | 0.244 | 0.272 | **0.240** | 0.272 | 0.258 | 0.278 | 0.265 | 0.286 | 0.251 | 0.294 | 0.268 | 0.321 | 0.265 | 0.315 |
| Exchange-Rate | 96 | 0.088 | **0.205** | 0.094 | 0.215 | 0.095 | 0.218 | **0.085** | **0.205** | 0.090 | 0.235 | 0.086 | 0.206 | 0.088 | **0.205** | 0.107 | 0.234 | 0.093 | 0.228 | 0.088 | 0.218 |
| | 192 | **0.176** | **0.299** | 0.177 | 0.301 | 0.228 | 0.339 | 0.177 | **0.299** | 0.187 | 0.343 | 0.177 | **0.299** | **0.176** | **0.299** | 0.226 | 0.344 | 0.185 | 0.332 | **0.176** | 0.315 |
| | 336 | **0.301** | **0.397** | 0.379 | 0.449 | 0.342 | 0.431 | 0.326 | 0.412 | 0.353 | 0.473 | 0.331 | 0.417 | **0.301** | **0.397** | 0.367 | 0.448 | 0.354 | 0.468 | 0.313 | 0.427 |
| | 720 | 0.917 | 0.720 | 0.924 | 0.726 | 0.848 | 0.712 | **0.837** | 0.690 | 0.934 | 0.761 | 0.847 | 0.691 | 0.901 | 0.714 | 0.964 | 0.746 | 0.872 | 0.727 | 0.839 | 0.695 |
| | avg | 0.371 | 0.405 | 0.393 | 0.423 | 0.378 | 0.425 | 0.356 | **0.401** | 0.391 | 0.453 | 0.360 | 0.403 | 0.367 | 0.404 | 0.416 | 0.443 | 0.376 | 0.439 | 0.354 | 0.414 |
| $1^{st}$ Count | | **20** | **24** | 3 | 2 | 0 | 0 | 5 | 8 | 3 | 1 | 0 | 1 | 2 | 3 | 1 | 0 | 0 | 0 | 2 | 0 |

- The Dual-Domain Cross-Attention mechanism demonstrates strong modeling capability for aligning frequency and time-domain information, particularly in capturing periodicity (e.g., with 15-minute intervals in ETTm1 and ETTm2 datasets).

- The contrastive Auxiliary loss exhibits varying degrees of impact across all datasets, benefiting from the adversarial interplay between positive-negative sample pairs and the competition among different augmentation strategies after introducing enhanced sequences.

## A.5.2 ABLATIONS ON AUGMENTATION STRATEGIES

We further discussed the contrastive output layer, augmented data type and contrastive output method of the contrastive auxiliary architecture.

In this ablation setting, the output positions of the encoder and projection are earlier and later than the predictor, respectively. Clearly, the output of the last layer, i.e., the projection layer, will be

Table 7: Ablations on adaptive convolution embedding, wavelet decomposition, dual-domain cross-attention and contrastive auxiliary loss (base and pairwise contrastive loss) in the four ETT-datasets long-term forecasting benchmarks with a fixed prediction length=96. Case 1 is the official design in FreqMixAttNet .

| Case | Adaptive convolution | Wavelet Decomposition | dual-domain cross-attention | contrastive auxiliary loss | | ETTh1 | | ETTh2 | | ETTm1 | | ETTm2 | |
|------|------|------|------|------|------|------|------|------|------|------|------|------|------|
| | | | | base auxiliary | pairwise | MSE | MAE | MSE | MAE | MSE | MAE | MSE | MAE |
| 1 | ✓ | ✓ | ✓ | ✓ | ✓ | 0.362 | 0.387 | 0.276 | 0.328 | 0.310 | 0.347 | 0.169 | 0.249 |
| 2 | ✓ | ✗ | ✓ | ✓ | ✓ | 0.363 | 0.389 | 0.280 | 0.331 | 0.311 | 0.349 | 0.173 | 0.254 |
| 3 | ✗ | ✓ | ✓ | ✓ | ✓ | 0.380 | 0.392 | 0.352 | 0.387 | 0.419 | 0.355 | 0.178 | 0.257 |
| 4 | ✓ | ✓ | ✗ | ✓ | ✓ | 0.378 | 0.388 | 0.288 | 0.335 | 0.343 | 0.363 | 0.182 | 0.259 |
| 5 | ✓ | ✓ | ✓ | ✗ | ✓ | 0.380 | 0.392 | 0.341 | 0.377 | 0.318 | 0.355 | 0.178 | 0.257 |
| 6 | ✓ | ✓ | ✓ | ✓ | ✗ | 0.374 | 0.390 | 0.287 | 0.336 | 0.327 | 0.358 | 0.176 | 0.255 |

subject to greater interference. This is because after compression from high-dimensional space to low-dimensional space, the information of positive and negative sample pairs becomes more unstable.

**contrastive output layer** To verify the effectiveness of different output layer of contrastive, we conducted a detailed ablation study by performing experiments, shown in table 8, including encoder, predict, and projection layer of form the sacle-specific outputs $h^a$.

Table 8: Ablations on contrastive output layer on the four ETT-datasets long-term forecasting benchmarks. 1 is the official design in FreqMixAttNet .

| Case | contrastive output layer | ETTh1 | | ETTh2 | | ETTm1 | | ETTm2 | |
|------|------|------|------|------|------|------|------|------|------|
| | | MSE | MAE | MSE | MAE | MSE | MAE | MSE | MAE |
| 1 | predicter | 0.362 | 0.387 | 0.276 | 0.328 | 0.310 | 0.347 | 0.169 | 0.249 |
| 2 | encoder | 0.363 | 0.388 | 0.285 | 0.331 | 0.312 | 0.349 | 0.172 | 0.252 |
| 3 | projection | 0.365 | 0.392 | 0.279 | 0.331 | 0.312 | 0.350 | 0.174 | 0.254 |

**Augmented data type** To verify the effectiveness of different type of augmented data, we conducted a detailed ablation study by performing experiments, shown in table 9, including freq-domain, time-domain and dual-domain mixing-up of augmented data.

The variations (1%–2.6%) of temporal, frequency-domain, and hybrid loss types on fine-grained datasets (ETTm*) are significantly greater than those on coarse-grained datasets (ETTh*) (0.03%–0.83%), fully demonstrating that frequency-domain loss enhances learning ability more effectively for datasets richer in frequency information.

Table 9: Ablations on Augmented data type on the four ETT-datasets long-term forecasting benchmarks. 1 is the official design in FreqMixAttNet .

| Case | Augmented data type | ETTh1 | | ETTh2 | | ETTm1 | | ETTm2 | |
|------|------|------|------|------|------|------|------|------|------|
| | | MSE | MAE | MSE | MAE | MSE | MAE | MSE | MAE |
| 1 | freq-domain | 0.362 | 0.387 | 0.276 | 0.328 | 0.310 | 0.347 | 0.169 | 0.249 |
| 2 | time-domain | 0.362 | 0.387 | 0.278 | 0.329 | 0.313 | 0.351 | 0.174 | 0.254 |
| 3 | mixing-up | 0.362 | 0.388 | 0.278 | 0.329 | 0.313 | 0.350 | 0.173 | 0.253 |

**contrastive output method** To verify the effectiveness of different method of contrastive output, we conducted a detailed ablation study by performing experiments, shown in table 10, including stack&sum and concat.

Table 10: Ablations on contrastive output method on the four ETT-datasets long-term forecasting benchmarks. 1 is the official design in FreqMixAttNet .

| Case | contrastive output method | ETTh1 | | ETTh2 | | ETTm1 | | ETTm2 | |
|------|---------------------------|-------|-------|-------|-------|-------|-------|-------|-------|
| | | MSE | MAE | MSE | MAE | MSE | MAE | MSE | MAE |
| 1 | stack&sum | 0.362 | 0.387 | 0.276 | 0.328 | 0.310 | 0.347 | 0.169 | 0.249 |
| 2 | concat | 0.364 | 0.387 | 0.280 | 0.330 | 0.310 | 0.348 | 0.176 | 0.256 |

## A.6 HYPERPARAMTER SENSITIVITY

### A.6.1 AUGMENT WEIGHT

we experimented the augment weight $\beta 1$ from 0.005, 0.01, 0.03, 0.04, 0.05, 0.06, 0.07, 0.08 for augment and predict length of 720 on the ETT datasets, to analyst the sensitivity of augment for super long-term forecasting, as illustrated in table 11.

Table 11: Multivariate long-term forecasting results with a unified lookback window L=96. The results are resulted from prediction lengths of 720. The best results are in bold and the second best are underlined.

| Models | 0.005 | | 0.01 | | 0.03 | | 0.04 | | 0.05 | | 0.06 | | 0.07 | | 0.08 | |
|--------|-------|-------|-------|-------|-------|-------|-------|-------|-------|-------|-------|-------|-------|-------|-------|-------|
| Metric | MSE | MAE | MSE | MAE | MSE | MAE | MSE | MAE | MSE | MAE | MSE | MAE | MSE | MAE | MSE | MAE |
| ETTh1 | 0.4529 | 0.4613 | 0.4517 | 0.4607 | 0.4526 | 0.4609 | 0.4527 | 0.4611 | 0.4511 | 0.4613 | 0.4510 | 0.4613 | 0.4517 | 0.4607 | 0.4516 | 0.4619 |
| ETTh2 | 0.4153 | 0.4349 | 0.4145 | 0.4344 | 0.4137 | 0.4339 | 0.4138 | 0.4339 | 0.4141 | 0.4340 | 0.4141 | 0.4340 | 0.4150 | 0.4347 | 0.4145 | 0.4343 |
| ETTm1 | 0.4450 | 0.4330 | 0.4439 | 0.4314 | 0.4453 | 0.4332 | 0.4457 | 0.4338 | 0.4445 | 0.4318 | 0.4440 | 0.4317 | 0.4434 | 0.4318 | 0.4433 | 0.4325 |
| ETTm2 | 0.3930 | 0.3920 | 0.3953 | 0.3932 | 0.3919 | 0.3908 | 0.3935 | 0.3922 | 0.3936 | 0.3917 | 0.3937 | 0.3913 | 0.3919 | 0.3898 | 0.3919 | 0.3901 |

### A.6.2 ALPHA

**Cross Prediction-Length Analysis**

From the perspective of stability affected by alpha, as the prediction length increases from 96 to 720, the variance (the standard deviation, STD) of both MSE and MAE increases by more than three times, as illustrated in table 12, indicating a deterioration in stability.

Table 12: Long-term forecasting performance comparison of different prediction lengths within the same dataset ETTh1. All input lengths are 96. A lower MAE, MAPE or RMSE indicates a better prediction.

| Models | 0.1 | | 0.3 | | 0.5 | | 0.7 | | 0.9 | | STD | |
|--------|-------|-------|-------|-------|-------|-------|-------|-------|-------|-------|-------|-------|
| prediction length | MSE | MAE | MSE | MAE | MSE | MAE | MSE | MAE | MSE | MAE | MSE | MAE |
| 96 | 0.3888 | 0.3624 | 0.3891 | 0.3629 | 0.3872 | 0.3611 | 0.3871 | 0.3617 | 0.3873 | 0.3619 | 0.10% | 0.07% |
| 720 | 0.4598 | 0.4652 | 0.4575 | 0.4649 | 0.4562 | 0.4642 | 0.4543 | 0.4627 | 0.4517 | 0.4607 | 0.31% | 0.19% |

**Cross-dataset Analysis**

we experimented the alpha value from 0.1, 0.3, 0.5, 0.7, 0.9 for augment and predict length of 720 on the ETT datasets, to analyst the sensitivity of alpha for super long-term forecasting, as illustrated in table 13.

Table 13: Long-term forecasting results in the ETT datasets with multiple variates. All input lengths are 96 and prediction lengths are 720. A lower MAE, MAPE or RMSE indicates a better prediction.

| Models | 0.1 | | 0.3 | | 0.5 | | 0.7 | | 0.9 | |
|--------|-----|-----|-----|-----|-----|-----|-----|-----|-----|-----|
| Metric | MSE | MAE | MSE | MAE | MSE | MAE | MSE | MAE | MSE | MAE |
| ETTh1 | 0.4598 | 0.4652 | 0.4575 | 0.4649 | 0.4562 | 0.4642 | 0.4543 | 0.4627 | 0.4517 | 0.4607 |
| ETTh2 | 0.4149 | 0.4348 | 0.4140 | 0.4342 | 0.4141 | 0.4342 | 0.4137 | 0.4339 | 0.4136 | 0.4338 |
| ETTm1 | 0.4433 | 0.4325 | 0.4485 | 0.4337 | 0.4480 | 0.4344 | 0.4493 | 0.4350 | 0.4487 | 0.4344 |
| ETTm2 | 0.3920 | 0.3899 | 0.3937 | 0.3915 | 0.3946 | 0.3926 | 0.3956 | 0.3932 | 0.3953 | 0.3938 |

### A.6.3 FULL HYPERPARAMETER

In addition, we conducted hyperparameter sensitivity analyses on the learning rate, dropout rate, $d_{model}$, number of attention heads ($n_{heads}$), and the two weighting coefficients in the contrastive learning loss, that is the value of $\beta2$ and $\beta3$ metioned in section 3.2.5. However, due to the substantial computational cost associated with varying prediction lengths, we fixed the prediction length at 96 and performed the following ablation studies on the same dataset, ETTh1, as illustrated in Figure 3 and 4, when augment weight increases, a significant "cliff-like" fluctuation in MSE occurs when its value ranges between 0.05 and 0.1. However, when the value exceeds 0.1, the MSE returns to a stable state. This indicates higher sensitivity of the parameter, likely caused by the non-stationary nature of frequency-domain information. For the augment Contrastive weight1 parameter, the performance of MAE gain a reduction at the thousandths place, but the MSE performance remains more stable. In contrast, changes in alpha and augment Contrastive weight2 result in stronger model robustness, with both MSE and MAE demonstrating robust stability performance. Therefore, we will adjust the combination of these four key hyper-parameters to trade off performance.

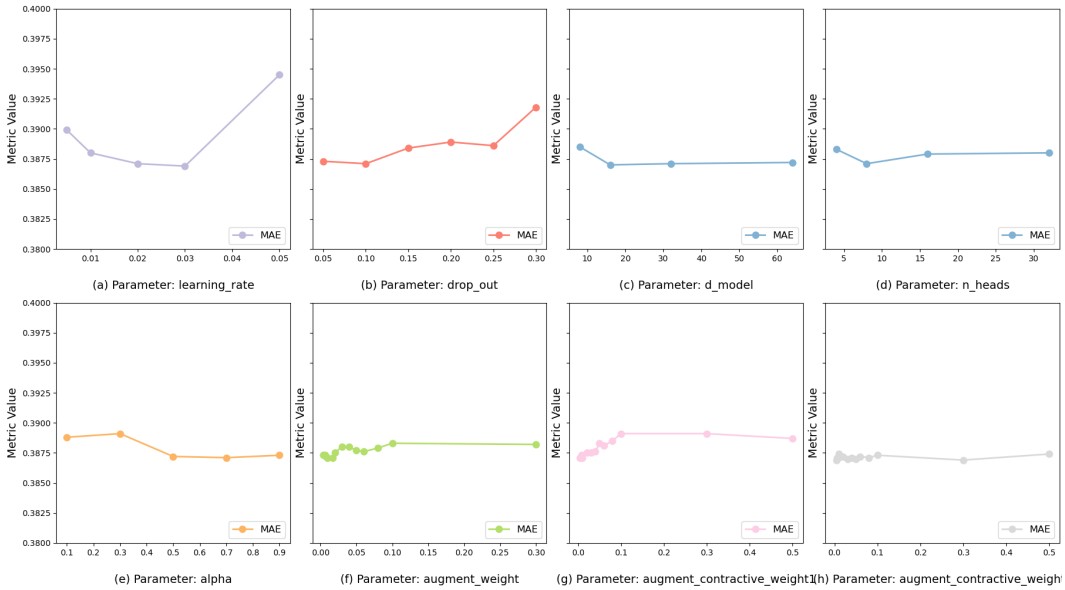

Figure 3: MAE Performance of Hyper-parameters Analysis.

### A.6.4 WHY FREQUENCY MIXING ENHANCES DOMAIN GENERALIZATION

To address the reviewer's concern regarding the theoretical justification of frequency mixing, we provide a principled explanation grounded in (i) spectral stability theory, (ii) standard domain-shift models, and (iii) invariant representation learning.

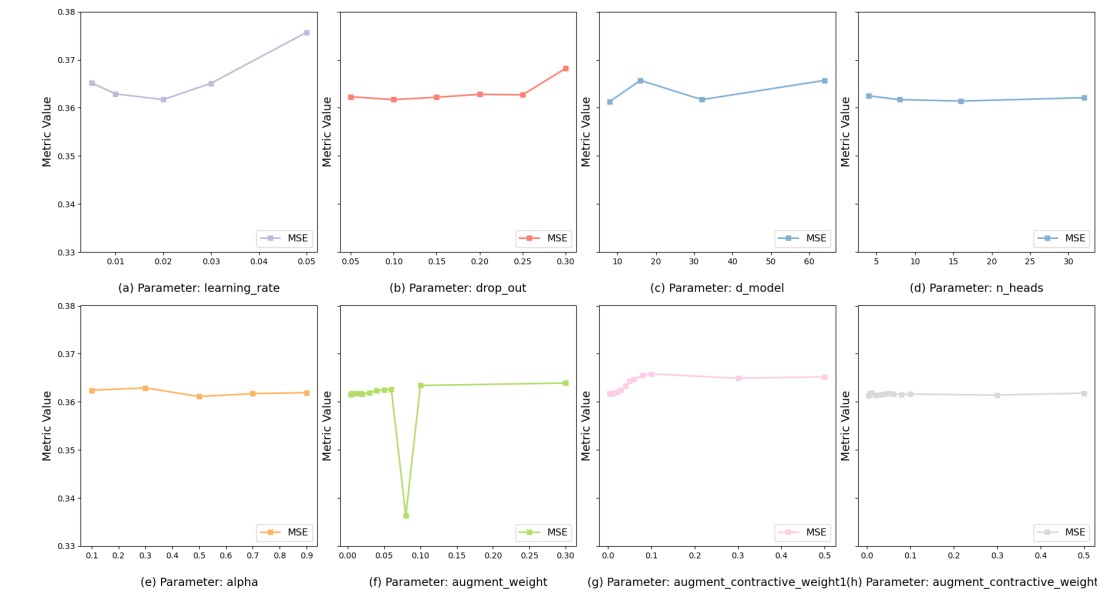

Figure 4: MSE Performance of Hyper-parameters Analysis.

**Spectral Contraction Under Domain Shifts**    A widely adopted formulation for domain generalization on time series models inter-domain variation as additive distortions:

$$x^{(d)}(t) = x(t) + \delta_d(t), \tag{20}$$

where $\delta_d(t)$ denotes domain-specific perturbations such as amplitude drift, jitter, offsets, and noise bursts.

Fourier stability theory shows that the $L_2$ deviations in the spectral domain induced by these perturbations are strictly smaller than those in the temporal domain:

$$\left\| \mathcal{F}(x) - \mathcal{F}(x^{(d)}) \right\|_2 \ll \left\| x - x^{(d)} \right\|_2. \tag{21}$$

In other words,

$$\delta_d(t)$$

may induce large pointwise distortions in the time domain but only minor variations in dominant frequency components. Eq. (21) implies that the spectral domain intrinsically *contracts* domain shift. Therefore, forcing seasonal representations into frequency space and mixing adjacent bands naturally amplifies domain-invariant spectral structures.

**Frequency Mixing as Learnable Spectral Filtering**    A complementary theoretical view originates from the convolution theorem. Applying a spectral filter $H(\omega)$ to a signal corresponds to a time-domain convolution with kernel $h(t)$:

$$Y(\omega) = H(\omega) X(\omega), \tag{22}$$
$$y(t) = (h * x)(t), \tag{23}$$

where $h(t) = \mathcal{F}^{-1}(H(\omega))$.

Our frequency mixing operation constructs a data-driven spectral filter by aggregating adjacent frequency bands:

$$\widetilde{X}_k = \alpha X_k + (1 - \alpha) X_{k \pm 1}, \tag{24}$$

which implicitly defines a learnable filtering kernel $H_{\text{mix}}(\omega)$ and a corresponding long-range convolution kernel $h_{\text{mix}}(t) = \mathcal{F}^{-1}(H_{\text{mix}}(\omega))$.

This yields two key consequences:

1. It acts as a *structured long-range convolution* that stabilizes global periodic components—precisely those least affected by domain shifts.

2. It suppresses high-frequency, domain-specific distortions (since averaging adjacent bands reduces spectral variance in unstable components).

In contrast, our time-domain branch (adaptive convolution or local attention) captures short-range, local dependencies. Combining Eqs. (23) and (24), the model yields a joint filter bank consisting of:

- global, spectrally defined convolution kernels stabilizing structure across domains, and
- local, context-adaptive temporal operators that respond to domain-specific fluctuations.

From the perspective of invariant representation learning, this hybrid filtering scheme increases the likelihood that domain-invariant periodic structures are preserved while domain-variant fluctuations are attenuated.

**Variance-Minimizing Feature Selection via Cross-Domain Attention** The benefit of spectral mixing is further strengthened by cross-domain attention:

$$\text{Attn}(Q_t, K_f, V_f), \tag{25}$$

where temporal features $Q_t$ query spectral representations $(K_f, V_f)$.

In the DG literature (e.g., Domain-Invariant Component Analysis), invariant features are characterized by minimal variance across domains. Let $\Sigma_d(\cdot)$ denote the covariance under domain $d$. Cross-domain attention implicitly performs:

$$\arg\min_\omega \ \text{Var}_d(X(\omega)), \tag{26}$$

i.e., selecting spectral components exhibiting minimal inter-domain variability.

Under this view:

- frequency mixing *constructs* more stable spectral regions by reducing high-frequency variance, and
- cross-domain attention *retrieves* these stable regions conditioned on temporal context.

Formally, combining Eqs. (21) and (26) indicates that frequency mixing + cross-domain attention jointly minimize domain-induced spectral variability:

$$\text{Var}_d\left(\widetilde{X}(\omega)\right) \ \leq \ \text{Var}_d(X(\omega)), \tag{27}$$

providing a principled explanation for the improved DG performance.

**Summary** Frequency mixing is theoretically supported on three pillars:

1. **Spectral contraction:** domain shifts shrink in the frequency domain (Eq. 21).

2. **Learnable filter design:** mixing adjacent bands corresponds to constructing global convolution kernels that suppress domain-specific distortions (Eqs. 22–24).

3. **Invariant retrieval:** cross-domain attention prioritizes minimal-variance spectral regions across domains (Eq. 26).

These insights jointly explain why frequency mixing enhances domain generalization performance.

