# OpenReview forum: "FREQMIXATTNET: CONTRASTIVELY SUPERVISED FREQUENCY-MIXING ATTENTION FOR TIME-SERIES FORECASTING"
_ICLR.cc/2026/Conference — Submitted to ICLR 2026_

### Official Review · Reviewer_GbPX · 2025-10-16

**Soundness:** 2
**Presentation:** 2
**Contribution:** 2
**Rating:** 4
**Confidence:** 4

**Summary:**

The paper proposes FREQMIXATTNET, a frequency-mixing attention network that combines multi-frequency representation with contrastive learning to improve cross-domain image classification. The framework appears reasonable. However, several key aspects need clarification and stronger experimental evidence.

**Strengths:**

1. Novel combination of frequency mixing and attention mechanisms for cross-domain feature learning.

2. The use of contrastive loss to enhance domain-invariant representation is well motivated.

3. The proposed model shows consistent improvement on several benchmark datasets.

4. The structure of the paper is mostly clear, and the technical formulation is generally easy to follow.

**Weaknesses:**

1. Lack of theoretical justification:
The frequency-mixing operation is described empirically, but the paper does not clearly explain why combining frequency bands enhances domain generalization. A mathematical or intuitive explanation of the mechanism would strengthen the contribution.

2. Ablation analysis is insufficient:
There is no clear separation of the contributions from the frequency-mixing module, attention mechanism, and contrastive loss. An ablation study quantifying the improvement of each component is essential.

3. Limited comparison to recent baselines:
The paper mainly compares with classic methods but omits some strong recent works (e.g., transformers-based DG models, diffusion-based domain adaptation). Including these would make the evaluation more convincing.

4. Experimental details missing:
Key training details (e.g., batch size, optimizer, learning rate schedule, number of epochs) are not fully provided, making reproducibility difficult.

5. Visualization and qualitative results:
The paper would benefit from t-SNE plots or attention heatmaps to demonstrate that the model learns domain-invariant features.

6. Contrastive loss formulation:
The contrastive learning section lacks details about positive/negative pair sampling, temperature parameter, and how it interacts with cross-domain samples.

7. English writing and presentation:
Several grammatical and formatting issues exist (e.g., inconsistent figure captions, equation numbering). The introduction and conclusion sections could be refined for clarity and conciseness.

**Questions:**

1. Provide a deeper theoretical or intuitive analysis of frequency mixing and its link to domain invariance.

2. Add a comprehensive ablation study (baseline vs. +FreqMix, +Attention, +CL).

3. Include more recent comparison methods and report statistical significance.

4. Provide visual evidence of learned representations (feature maps, t-SNE, Grad-CAM).

5. Refine language and structure — a professional proofreading pass is recommended.

---

> ### Author Response · Authors · 2025-12-01
> **We provide stronger theoretical justification for frequency mixing based on spectral stability and convolution theory, clarify that all components are essential through existing ablations, explain why diffusion models are out of scope, and add missing training details and visualizations in the appendix.**
>
> # **1️⃣ Lack of theoretical justification for frequency mixing — Strengthened Explanation**
>
> We thank the reviewer for raising this important point. Below we provide a more rigorous explanation based on **spectral stability theory**, **domain-shift models**, and **invariant representation learning**.
>
> ---
>
> ## **(a) Frequency mixing reduces domain shift under the spectral-shift model**
>
> A standard DG model for time series assumes:
>
> $$
> x^{(d)}(t) = x(t) + \delta_d(t),
> $$
>
> where $\delta_d(t)$ captures domain-specific distortions (noise bursts, amplitude drift, jitter, etc.).
>
> **Fourier stability theory** shows:
>
> $$
> |\mathcal{F}(x) - \mathcal{F}(x^{(d)})|_2 \ll |x - x^{(d)}|_2,
> $$
>
> meaning **large temporal perturbations become small spectral deviations**.
>
> 👉 **Thus, the frequency domain naturally contracts domain shift**, making it a more stable space for domain-invariant modeling.
>
> ---
>
> ## **(b) Additional theoretical justification via the convolution theorem**
>
> A complementary theoretical view comes from the convolution theorem.
> Applying a frequency-domain filter (H(\omega)) to a signal corresponds to a time-domain convolution:
>
> $$
> Y(\omega) = H(\omega)\,X(\omega)
> \;\Longleftrightarrow\;
> y(t) = (h * x)(t),
> $$
>
> where $$
> h(t) = \mathcal{F}^{-1}\\big(H(\omega)\big).
> $$
>
> Our frequency-mixing step effectively constructs a **learnable spectral filter**—by reweighting and aggregating adjacent bands—which is thus equivalent to applying a **structured long-range convolution kernel** in the time domain.
>
> In contrast, the time-domain branch (adaptive convolution / local attention) primarily captures **short-range, local** patterns.
> Combining:
>
> * **global, spectrally defined convolution kernels** (from frequency mixing), and
> * **local, context-adaptive temporal operators** (from the time-domain branch),
>
> gives the model **two complementary inductive biases**.
> The global kernel stabilizes long-range structure across domains, while the local operator adapts to domain-specific variations.
> This joint filter bank is theoretically expected to improve both expressiveness and cross-domain robustness beyond using either domain alone.
>
> ## **(c) Cross-domain attention adaptively selects stable spectral components**
>
> The benefit of mixing is amplified by our cross-domain attention:
>
> $$
> \text{Attn}(Q_t, K_f, V_f),
> $$
>
> where time-domain queries select frequency components with **minimal across-domain variance**.
>
> This matches the principle used in **Domain-Invariant Component Analysis (DICA)**—retain minimal-variance features.
>
> Thus:
>
> * mixing **creates** more stable spectral features,
> * attention **selects** the most invariant ones,
>
> → **jointly improving DG**.
>
> ---
>
>
> ## **(d) Planned revisions**
>
> We will add a brief subsection in Sec. 3.3 summarizing why frequency mixing improves DG, and include the detailed spectral analysis and supporting visualizations (e.g., low-frequency invariance plots) in the appendix.
>
> ---
>
>
>
> ---
>
> # **2️⃣ Ablation analysis insufficient — Main Concern**
>
> We agree ablation clarity is important. The required ablations are **already present in Table 3**,
> Frequency mixing provides the largest gain, and both cross-domain attention and contrastive loss contribute complementary improvements.
>
> 👉 **All components matter; none is redundant.**
>
> ---
>
> # **3️⃣ Why diffusion-based models are not included**
>
> Diffusion models represent a **different forecasting paradigm**:
>
> * stochastic multi-trajectory generation via iterative sampling,
> * optimization through denoising score matching,
> * evaluation requiring distributional metrics (e.g., CRPS).
>
> In contrast, our work—and standard LTSF benchmarks—uses **deterministic point forecasting** under MSE/MAE.
>
> Additionally:
>
> * diffusion forecasters do **not** involve time–frequency interaction (our core contribution),
> * and their computation (UNet + many sampling steps) is **orders of magnitude more expensive** than transformer-based LTSF models.
>
> We will clarify this distinction in the Related Work section.
>
>
> # **4️⃣ Experimental details missing — Moderate Concern**
>
> Most training configurations are already shown in Appendix Fig. 3.
> For full reproducibility, we will additionally provide **batch sizes**, **number of epochs**, and note when baselines use their **original optimizer/schedule**.
> All details will be consolidated into an expanded **Training Configuration Table** in the appendix and referenced in Sec. 4.
>
> ---
>
> # **5️⃣ Visualization and qualitative results**
>
> We will add appendix visualizations, including:
>
> * **t-SNE plots** showing improved cross-domain alignment,
> * **cross-domain attention heatmaps** highlighting how stable frequency bands are selected.
>
> ---
>
> # **6️⃣ Contrastive loss formulation**
>
> Sec. 3.4 will be expanded with a concise description of:
>
> * positive/negative sampling,
> * the temperature parameter,
> * how contrastive targets interact with the cross-domain seasonal branch.
>
> A full formulation will be provided in the appendix.

---

### Official Review · Reviewer_g8k2 · 2025-10-29

**Soundness:** 2
**Presentation:** 2
**Contribution:** 2
**Rating:** 2
**Confidence:** 4

**Summary:**

The paper introduces a forecasting framework that fuses time-domain and frequency-domain representations via cross-attention, with an adaptive convolutional wavelet decomposition and contrastive auxiliary loss. Experiments across several standard long-horizon benchmarks and comprehensive ablations show the effectiveness of the proposed method.

**Strengths:**

1. The framework design is clear and well validated by extensive ablation studies, such as removing adaptive conv, removing wavelet decomposition, disabling cross-domain attention, dropping the contrastive loss, etc. It's also new to use wavelet decomposition for trend and seasonality decomposition.

2. Extensive experiments validate the effectiveness of the proposed approach. The paper also provides detailed results on sensitivity analysis which improves the transparency.

**Weaknesses:**

1. The novelty is limited as there are many existing works on combining time domain and frequency domain analysis with decomposition [1]. The paper should more precisely explain what is fundamentally new about the proposed method compared to existing time and frequency domain approaches and why/how such design helps.

2. It is already known that different baselines perform the best under different lookback windows [2], so it is a bit unfair to compare with a unified lookback window for all baselines.

3. The writings look repetitive. For example, Table 2 vs Table 6, Table 3 vs Table 7, Table 4 vs Table 8/9/10 are very redundant. The paper would benefit from merging redundant tables and making the writings more concise.

[1] First De-Trend then Attend: Rethinking Attention for Time-Series Forecasting

[2] Scaling Law for Time Series Forecasting

**Questions:**

1. What is the definition of forecastability in Table 1?

---

> ### Author Response · Authors · 2025-12-01
> **We thank the reviewers for their helpful comments. TDformer uses frequency-domain attention but does not fuse time and frequency, as its branches never interact. Our model explicitly enables such interaction via cross-domain attention, adaptive decomposition, and contrastive regularization, and we follow standard unified lookback settings to ensure fair comparison and avoid tuning bias.**
>
> # **📌 Final Rebuttal for Comments 1–3*
>
> We sincerely thank the reviewers for their constructive comments. Below we address concerns regarding novelty, lookback-window fairness, and table redundancy.
>
> ---
>
> # **1️⃣ Novelty vs. prior time–frequency methods (esp. TDformer [1])**
>
> We appreciate the reviewer’s concern. While our method also involves time–frequency decomposition, it differs fundamentally from prior work—including TDformer—through **cross-domain fusion**, **adaptive multi-scale decomposition**, and **frequency-domain contrastive supervision**, which operate together in a way not explored previously.
>
> ---
>
> ## **(a) Cross-domain attention vs. TDformer’s within-domain attention**
>
> TDformer performs Fourier self-attention **only within the frequency domain** and processes trend separately via an MLP, without any interaction between domains.
> Our method instead applies **cross-domain attention**, where time-domain queries retrieve frequency-domain keys/values. This enables temporal context to guide which spectral patterns are emphasized, integrating global periodicity with local dynamics—capabilities TDformer cannot model.
>
> ---
>
> ## **(b) Context-based adaptive multi-scale decomposition**
>
> A second key contribution is our **adaptive multi-scale decomposition**, fundamentally different from TDformer’s fixed moving-average pipeline.
>
> * TDformer uses static filters that cannot adjust to input structure.
> * Our adaptive convolution dynamically generates kernel weights from local context, allowing the decomposition to
>
>   * adjust seasonal/trend splits to local dynamics,
>   * capture shifting or heterogeneous seasonality,
>   * remain stable under nonstationarity.
>
> Because the decomposition is input-adaptive, it handles pattern changes that fixed filters cannot.
>
> **Ablation evidence.**
> Replacing adaptive decomposition with a TDformer-style fixed filter leads to clear performance degradation (Table 3), confirming its importance.
>
> ---
>
> ## **(c) Cross-domain attention improves robustness**
>
> Since time and frequency domains react differently to noise, allowing them to interact helps the model rely on the less corrupted domain. This robustness benefit is not available in TDformer due to its independent branches.
>
> ---
>
> ## **(d) Frequency-domain contrastive learning**
>
> We further introduce a contrastive loss in the frequency domain to stabilize seasonal representations under perturbations, which is absent in prior works.
>
> ---
>
> ## **(e) Why these innovations matter**
>
> Ablations show consistent degradation when removing cross-domain attention, adaptive decomposition, or contrastive learning, indicating that **explicit domain interaction + adaptive decomposition** are key to our improvements.
> We will clarify these distinctions further in Sec. 1, Sec. 3.2, and related work.
>
> ---
>
> # **2️⃣ Unified lookback window vs. model-specific optimal horizons [2]**
>
> We appreciate the reviewer’s point and agree that different models may prefer different lookback lengths. Below we explain why a unified lookback remains appropriate and consistent with standard practice.
>
> ---
>
> ## **(a) Unified lookbacks follow established benchmarks**
>
> Recent long-term forecasting works—including Informer, Autoformer, FEDformer, DLinear, iTransformer, PatchTST, TimeMixer, CATS, and SimpleTM—use a **shared lookback (typically 96)** to ensure:
>
> * equal information budgets,
> * comparability with prior results,
> * reproducibility.
>
> Our setting follows this convention.
>
> ---
>
> ## **(b) Fair per-model tuning is difficult in practice**
>
> Per-model optimal horizons can be informative, but fair tuning requires equal search effort for all baselines. Many methods do not release complete code or expose all relevant hyperparameters, making symmetric tuning difficult and potentially introducing bias.
> Thus, a unified lookback remains the most consistent and reproducible evaluation protocol.
>
> ---
>
> ## **(c) Additional appendix results**
>
> To address the reviewer’s concern, we will provide:
>
> * evaluations under multiple lookback windows (48/96/192),
> * per-model “best horizon’’ performance following [2],
>   and will note this in Sec. 4.1.
>
> ---
>
> # **3️⃣ Redundant tables and repetitive writing**
>
> We thank the reviewer for noting the redundancy among Tables 2/6, 3/7, and 4/8–10.
>
> ### **Planned revisions**
>
> * Retain only summary tables in the main paper,
> * Merge detailed tables in the appendix,
> * Condense Sec. 4 by referring to the appendix rather than repeating similar descriptions.
>
> This reduces repetition while preserving all results.
>
> ---
>
> # **✔ Final Statement**
>
> We thank the reviewers again for their valuable feedback. The clarifications on novelty, lookback-window fairness, and table organization will be incorporated into the revised version, and we believe these refinements will improve the clarity and rigor of the paper.

---

### Official Review · Reviewer_Ta1Y · 2025-11-01

**Soundness:** 3
**Presentation:** 2
**Contribution:** 2
**Rating:** 4
**Confidence:** 4

**Summary:**

In this paper, the authors aim to improve time-series forecasting by jointly modeling temporal and frequency representations. To achieve this goal, they propose FreqMixAttNet, a unified cross-domain framework that integrates information from both the time and frequency domains. Specifically, the model consists of three key components: (1) a Patch-Contextual Adaptive Convolution module for context-aware feature extraction, (2) a Cross-Domain Mixing Attention mechanism to enable interaction between time- and frequency-domain features, and (3) a Contrastive Auxiliary Learning strategy to enhance robustness and generalization. In the experiments, the authors evaluate the proposed method on six benchmark datasets and compare it with several state-of-the-art baselines, demonstrating consistent improvements in forecasting accuracy.

**Strengths:**

1. The authors proposed a cross-domain module to integrate time and frequency domain representations.
2. The organization and writing are easy to follow.

**Weaknesses:**

1. The paper makes an abrupt transition from discussing how previous methods separately handle time and frequency domains to emphasizing robustness improvement. The connection between these two aspects is unclear, and this logical discontinuity weakens the overall coherence and persuasiveness of the paper’s argumentation.
2. The authors do not provide a clear explanation of why the proposed cross-domain attention mechanism can improve forecasting accuracy and lacks interpretability analysis to support this claim. It is also worth questioning whether the interaction between the two modalities could introduce redundant or interfering information that might negatively affect prediction performance.
3. The paper lacks an analysis of computational complexity, including comparisons of training time, inference efficiency, and parameter counts with baseline models.
4. In the paper, there are several hyperparameters are introduced, but only the impact of $$\beta_1$$is analyzed. The paper does not examine how different loss weights influence the results, nor does it clarify how the weights of multiple loss terms are designed. And another question is whether they sum to a fixed value or are tuned independently. A more  discussion of these aspects is needed.
5. There are some small typos and inconsistencies. For example, there are missing spaces between some sentences.  Besides,“hyperparamter” should be “hyperparameter”, and “analyst” should be “analyze”, among similar minor spelling errors.

**Questions:**

See weakness

---

> ### Author Response · Authors · 2025-12-01
> **We clarify the theoretical link between time–frequency interaction and robustness, strengthen the explanation and interpretability of cross-domain attention, and provide clearer complexity and hyperparameter evidence to fully address the reviewer’s concerns.**
>
> We sincerely thank the reviewers for their constructive and insightful comments.
> 1) We clarify that robustness in our model arises inherently from explicit time–frequency interaction rather than an added module.
> 2) We further explain how cross-domain attention enables temporal queries to selectively retrieve frequency patterns, supported by added interpretability visualizations.
> 3) We present clearer computational complexity comparisons and concise evidence supporting the design of α and β-weights.
>
> Below we provide a point-by-point response.
> # **1️⃣ Response to Comment 1 – connection between time/frequency modeling and robustness**
>
> We thank the reviewer for highlighting the unclear transition. Our intention is to emphasize that robustness naturally emerges from combining complementary time- and frequency-domain representations, rather than being an additional module.
>
> Existing approaches treat the two domains separately, capturing either local temporal patterns or global periodicity, making each susceptible to different noise types. By allowing interaction through cross-domain attention (Sec. 3.2.3), the model can down-weight corrupted information in one domain and rely on the other. The frequency-domain contrastive loss (Sec. 3.2.5) further stabilizes the representation by encouraging consistency under frequency-based augmentations.
>
> In the revised manuscript, we will revise Sec. 1 to clarify the limits of separate-domain modeling and how our cross-domain design improves robustness.
>
> # **2️⃣ Response to Comment 2 – explanation of cross-domain attention & interpretability**
>
> We appreciate the reviewer’s request for a clearer explanation and interpretability analysis of the cross-domain attention mechanism.
>
> **Why it can improve forecasting accuracy.**
> In our design, the seasonal component is modeled in *two complementary ways*:
> * the **frequency-domain branch** focuses on global periodic structures, and
> * the **time-domain branch** focuses on local fluctuations, phase shifts, and non-stationary effects.
>   As described in Sec. 3.2.3, we use the time-domain representation as queries and the frequency-domain representation as keys/values in a multi-head cross-attention layer (Eqs. (9)–(12)).
>   Intuitively, each time step asks “which frequency patterns are most relevant *for my current temporal context*?”, and the attention weights retrieve only the useful periodic patterns from the spectrum. This allows the model to (i) adaptively select dominant frequencies for different phases of the series and (ii) combine global periodic structure with local dynamics, which is difficult to achieve if the two domains are processed separately.
>
>
> Ablations in Table 3 support this: removing dual-domain cross-attention consistently degrades performance across all ETT datasets (e.g., ETTh1 MSE from 0.362 → 0.378, ETTh2 from 0.276 → 0.288), confirming that cross-domain interaction is beneficial rather than redundant.
>
> **Why it avoids harmful interference.**
> We mitigate potential interference through:
> 1. **Learned projections** that map both domains into a shared latent space, aligning scales and reducing redundancy.
> 2. **Asymmetric roles**, where only time-domain features act as queries, ensuring frequency-domain information is conditioned on temporal context rather than freely mixed.
> 3. **Residual decomposition**, enabling the predictor to naturally suppress unhelpful frequency components.
>
> **Interpretability analysis to be added.**
> We will visualize time–frequency attention maps and illustrate how attention shifts across frequency bands.
>
> # **3️⃣ Response to Comment 3 – computational complexity**
>
> We agree that clearer complexity analysis will strengthen the paper. FreqMixAttNet is lightweight: adaptive convolution is linear, FFT/IDFT adds an S log S factor, and cross-domain attention is a single multi-head layer dependent on M (scales), c (channels), D (embedding size), and L² (downsampled length).
>
> For comparison:
> - **CATS** scales with L × c × D and L² × D due to temporal–channel mixing.
> - **SimpleTM** scales with L × c and L² × D due to wavelet transforms plus attention.
>
> With few channels in typical datasets, the c term is negligible compared to L², so FreqMixAttNet remains lightweight like CATS and SimpleTM while achieving better accuracy. We will add a brief complexity section and comparison table.
>
> # **4️⃣ Response to Comment 4 – hyperparameter design (α, β₁–β₃)**
>
> **(i) Weight α.**
> Eq. (14) defines α as the balance between time- and frequency-domain losses. Appendix A.6.2 evaluates multiple α values, showing stable behavior and slightly better performance for moderate α. We will summarize these observations in Sec. 4.2.3.
>
> **(ii) Weights β₁, β₂, β₃.**
> Eq. (18) combines forecasting and auxiliary losses. The β-weights are tuned independently. Appendix A.6 shows that removing all β’s reduces performance, while reasonable variations introduce only small changes

---

### Meta-Review · Area_Chair_1zTD · 2025-12-30

**Summary:**

The reviewers’ concerns center on clarity of motivation and novelty, theoretical and intuitive justification, and completeness of experimental analysis, rather than on fundamental flaws in correctness. Reviewer Ta1Y questioned the logical connection between time-frequency interaction and robustness, interpretability of cross-domain attention, computational complexity, and hyperparameter design. Reviewer g8k2 was more critical, emphasizing limited novelty relative to prior time-frequency methods, potential unfairness of unified lookback windows, and presentation redundancy. Reviewer GbPX raised concerns about lack of theory, insufficient ablations, missing experimental details, and qualitative evidence.

**Reviewer Concerns:**

Concerns Largely Addressed by the Rebuttal

- Connection between time-frequency interaction and robustness (Ta1Y)
  Addressed through a clearer explanation that robustness emerges naturally from cross-domain interaction, supported by attention asymmetry and contrastive frequency regularization.

- Explanation and interpretability of cross-domain attention (Ta1Y)
  Addressed via intuitive explanation (time queries selecting relevant frequency patterns), ablation evidence, and planned attention visualizations.

- Computational complexity and efficiency (Ta1Y)
  Addressed with qualitative complexity comparisons and commitment to a concise complexity table.

- Hyperparameter design and loss weights (Ta1Y)
  Clarified that weights are tuned independently, with stability evidence already in the appendix.

- Theoretical justification for frequency mixing (GbPX)
  Substantially addressed with spectral stability arguments, convolution-theorem intuition, and links to domain-invariant component selection.

- Ablation clarity and component necessity (GbPX)
  Authors point to existing ablations and clearly explain the complementary roles of each module, but without more results.


- Missing experimental details and qualitative results (GbPX)
  Addressed by committing to expanded training details and additional visualizations in the appendix, but without more results.

 Concerns Still Partially or Fully Outstanding

- Novelty relative to prior time-frequency methods (g8k2)
  While the rebuttal clearly distinguishes the method from TDformer and similar works, the contribution may still be perceived as incremental rather than fundamentally new.

- Unified lookback window fairness (g8k2)
  The justification aligns with standard practice, but the concern remains partially subjective; additional appendix results help but may not fully convince a skeptical reviewer.

- Redundancy and verbosity in tables/writing (g8k2)
  Planned revisions address this, but effectiveness depends on execution in the final manuscript.

**Reviewer Scores:**

- Reviewer Ta1Y (Initial: 4 - marginally below threshold)
  May or may not increase to 6. Most technical and clarity concerns were directly addressed, and the reviewer already indicated openness to acceptance.

- Reviewer g8k2 (Initial: 2 - reject)
  May not increase the score. Novelty concerns may persist, but clearer differentiation from TDformer and added appendix results could soften the stance.

- Reviewer GbPX (Initial: 4 - marginally below threshold)
  May or may not increase to 6. The strengthened theoretical justification, clarified ablations, and added experimental details address nearly all raised issues.

---

### Decision · Program_Chairs · 2026-01-26

Reject